# SAME PRE-TRAINING LOSS, BETTER DOWNSTREAM: IMPLICIT BIAS MATTERS FOR LANGUAGE MODELS

## ABSTRACT

Language modeling on large-scale datasets leads to impressive performance gains on various downstream language tasks. The (validation) pre-training loss (or perplexity in autoregressive language modeling) is often used as the evaluation metric when developing language models since the pre-training loss tends to be well-correlated with downstream performance (which is itself difficult to evaluate comprehensively). Contrary to this conventional wisdom, this paper shows that 1) pre-training loss cannot fully explain downstream performance and 2) flatness of the model is well-correlated with downstream performance where pre-training loss is not. On simplified datasets, we identify three ways to produce models with the same (statistically optimal) pre-training loss but different downstream performance: continue pre-training after convergence, increasing the model size, and changing the training algorithm. These experiments demonstrate the existence of implicit bias of pre-training algorithms/optimizers—among models with the same minimal pre-training loss, they implicitly prefer more transferable ones. Toward understanding this implicit bias, we prove that SGD with standard mini-batch noise implicitly prefers flatter minima in language models, and empirically observe a strong correlation between flatness and downstream performance among models with the same minimal pre-training loss. We also prove in a synthetic language setting that among the models with the minimal pre-training loss, the flattest model transfers to downstream tasks.

## 1 INTRODUCTION

Large language models (LLMs) trained on internet-scale data have improved performance on a wide array of downstream tasks (Devlin et al., 2018; Yang et al., 2019; Radford et al., 2019; Raffel et al., 2020; Brown et al., 2020). These models are trained with a language modeling pre-training loss to "fill in the blanks"—either predicting the next token/word (autoregressive language modeling loss, or perplexity) or masked tokens (masked language modeling (MLM) loss).

In common practice, the *validation* pre-training loss is used to monitor the training process (Brown et al., 2020; Zhang et al., 2022a) and compare different models since the pre-training loss is generally strongly correlated with downstream performance (Hernandez et al., 2021). Moreover, theoretical works on understanding LLMs also focus on how the pre-training loss affects downstream performance. Saunshi et al. (2020); Wei et al. (2021); Xie et al. (2021) show that good pre-training loss, or fitting the language modeling conditional probability well, is a main reason for downstream success of LLMs. Their analyses generally treat the language models as blackboxes and do not take into account *how* the models represents the conditional probability.

In this paper, we question the conventional wisdom on the correlation between the validation pre-training loss and downstream performance for language modeling. Recent works have demonstrated that models with different architectures may have the same pre-training loss but different performance (Saunshi et al., 2022; Tay et al., 2021). Due to the expressivity of modern neural nets, many parameter configurations even within the *same* architecture can still have the same pre-training loss. A priori, it is unclear why all these configurations should have the same downstream performance.

We find that different parameter configurations with the same pre-training loss can indeed have different downstream performance, especially when the pre-training loss reaches a near-optimal level. Concretely, using simplified text datasets, we find three situations that demonstrate such a phenomenon:

- Even after the pre-training loss converges, models at a later time step still tend to perform better.

- Models trained by standard algorithms have better performance than adversarially trained models with the same pre-training loss.

- Larger models tend to perform better downstream than smaller models even if they have the same pre-training loss.

These situations are most prominent in the *saturation regime*, where the models are close to the minimal possible pre-training loss (aka the entropy of the conditional probability, which can be estimated in our simplified datasets). In the saturation regime, the pre-training loss of all models are almost the same, but the transferability to downstream tasks varies. Interestingly, this phenomenon also holds when linear probing on contextualized presentations is used for evaluating downstream performance instead of finetuning. Thus, even though the predicted conditional probabilities of two models are the same (and correct), the contextualized representations can behave differently.

In each of the first two cases above, we find two models with the same pre-training loss and the same architecture; but one has a better performance than the other. They only differ by the training algorithms that are used to produce them. Therefore, this suggests the training algorithms have an *implicit bias* toward one of these models—standard algorithms with more training steps biases towards parameter configurations that transfer better to downstream tasks. The third case has a more subtle but similar interpretation. There exists a hypothetical large model that represents the smaller model with worse performance (by padding zeros to the smaller model). The training algorithm on the large architecture could have chosen it, but did not. This suggests the algorithm has an implicit bias against the hypothetical model (which has an equally good loss).

In supervised settings, optimizers are known to have an implicit bias toward selecting generalizable models among all models with small *empirical* loss. E.g., see Damian et al. (2021); Li et al. (2021), which show that SGD implicitly biases toward flatter minima, and references therein. However, the role of implicit bias in self-supervised learning has not been studied and is conceptually different. Unlike in supervised learning, the gap between empirical and population self-supervised losses is typically small, and thus implicit bias does *not* seem to contribute to bridging this gap. Instead, the implicit bias selects local minima of the *population* self-supervised loss that transfer better to downstream tasks.

Why do the algorithms bias toward some type of models? In Section 3, we provide a first-cut theoretical analysis of the implicit bias in language modeling. Fortunately, despite the conceptual differences, mathematical tools from supervised settings can be straightforwardly adapted to language modeling settings. We prove that mini-batch SGD prefers flatter minima of population pre-training loss among all minima in the saturation regime. Interestingly, we obtain cleaner theoretical results for the standard mini-batch SGD, without the artificial label noise introduced in prior works (Damian et al., 2021; Li et al., 2021), partly because the mini-batch noise for LLMs does not vanish even at convergence.

We corroborate our theory with empirical evidence in Section 4. We show that for models with the same pre-training loss in the three situations above, flatness of the model (measured by the trace of Hessian of the loss, as predicted by the theory) strongly correlates with the downstream performance.

Finally, to complement the theory and experiments above, we also rigorously formalize the connection between flatness and downstream performance in a simplified Dyck language setting in Section 5. In this setting, we prove that there are many models with good MLM pre-training loss; among them, the flattest model learns the most useful features for downstream tasks. Here, results from the supervised setting cannot be readily adapted since they are obtained (partially) via generalization bounds (Wei & Ma, 2019a;b), which do not apply to the language modeling setting where the implicit bias is not related to the gap between the empirical and population loss. Proving the correlation between flatness and downstream performance in more general settings likely requires highly non-trivial and novel theoretical tools, and we hope to motivate future work on this topic.

## 2    THE EXISTENCE OF IMPLICIT BIAS IN LANGUAGE MODELING

In this section, we systematically investigate the relationship between pre-training loss and downstream performance with experiments. We find out that models with the same pre-training loss but different training procedures can have different downstream performance.

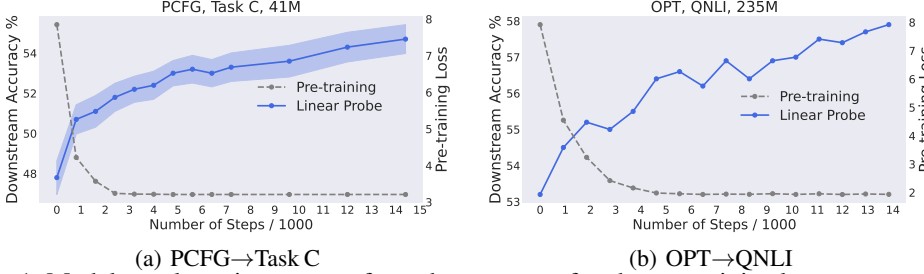

(a) PCFG→Task C            (b) OPT→QNLI

Figure 1: Models at a later time step performs better, even after the pre-training loss converges. (a) A model with 41M parameters pre-trained on the PCFG-generated dataset, and evaluated on task C. (b) A model with 235M parameters pre-trained on the OPT-generated dataset, and evaluated on QNLI.

## 2.1 FORMULATIONS

**Masked language modeling.** Consider a vocabulary $W = \{0, 1, ..., c\}$, where $0$ is a special token for the mask. Let $x = [x_1, ..., x_T]$ denote the input sequence of length $T$, and $x_{-t} = [x_1, ..., x_{t-1}, 0, x_{t+1}, ..., x_T]$ denote the masked sentence, where $t$ is sampled uniformly randomly and independently from $[T]$.[1] The MLM conditional probability refers to the probability of $x_t$ given the rest of the sequence $\Pr(x_t | x_{-t})$. We use $\Pr(\cdot | x_{-t})$ to denote the $c$-dimensional probability vector $\Pr(\cdot | x_{-t}) := [\Pr(x_t = 1 | x_{-t}), ..., \Pr(x_t = c | x_{-t})] \in \mathbb{R}^c$. In MLM pre-training, the model $f_\theta(\cdot)$ (parameterized by $\theta$) outputs the predicted MLM conditional probability vector $f_\theta(x_{-t}) \in \mathbb{R}^c$. The model is trained to predict the masked token $x_t$ given the rest of the sentence $x_{-t}$ with cross entropy loss, $L(\theta) = \mathbb{E}_{x,t}[\ell(f_\theta(x_{-t}), x_t)] = \mathbb{E}_{x,t}[-\log([f_\theta(x_{-t})]_{x_t})]$.

**Downstream evaluation.** The language model $f_\theta$ is composed of a feature extractor $h_\psi$, which outputs a sequence of contextual representations, and a linear classifier that outputs the conditional probability at every position. On downstream tasks, we use a randomly initialized $g_\phi$ on top of the pre-trained $h_\psi$. In fine-tuning, both $g_\phi$ and $h_\psi$ are trained, while in linear probe, only $g_\phi$ is updated. For fine-tuning, we use the contextual representations of the `cls` token. For linear probe, we concatenate the contextual representations of all the tokens together.

**Saturation regime.** To study models with the same pre-training loss, we introduce the saturation regime in this paper, where the model output equals the true conditional probability, $f_\theta(x_{-t}) = \Pr(\cdot | x_{-t})$. In the saturation regime, the MLM loss is equal to the entropy of the true conditional probability $L(\theta) = \mathbb{E}_{x,t}[-\log(\Pr(x_t | x_{-t}))] = \frac{1}{T}\sum_{t=1}^{T} H(x_t | x_{-t})$, which is also the optimal pre-training loss. Thus, *all* models in the saturation regime have the same, optimal pre-training loss, and we will show that they behave differently on downstream tasks. Our experiments use expressive enough architectures such that there are multiple parameter configurations in the saturation regime for our simplified datasets. For real large-scale data, it is currently computationally challenging to arrive at the saturation regime. However, we hope that our experiments can provide insights for even larger models in the future and for other regimes where pre-training loss does not explain downstream performance.

## 2.2 EXPERIMENTAL SETUP

We design controlled experiments to study the correlation between pre-training loss and downstream performance. In particular, we will find a set of models with almost the same pre-training loss. We effectively use the same architecture family so that the main difference between the models only stems from training algorithms. More details are provided in Section A.

**Datasets.** We introduce three generative models to produce simplified datasets, with which we can study various factors systematically. With the knowledge of the true generative models that generate the data, we can compute the true conditional probability and scale up the models until they approach the saturation regime to ensure they have almost the same pre-training loss. Moreover, we can generate unlimited amount of text for pre-training to avoid overfitting to the *empirical* pre-training loss.

1) *PCFG*-generated dataset. PCFG (Chomsky, 1956) generates sentences with probabilistic trees and is widely used to understand natural language (Johnson, 1998; Roark & Bacchiani, 2003; Kim et al., 2019; Yang et al., 2021). We randomly generate the production rules which satisfy the Chomsky Normal Form (Chomsky, 1956). The non-terminal symbols in the parse tree can be viewed as intrinsic

---

[1]For simplicity, we only consider masking out one one token in each sentence.

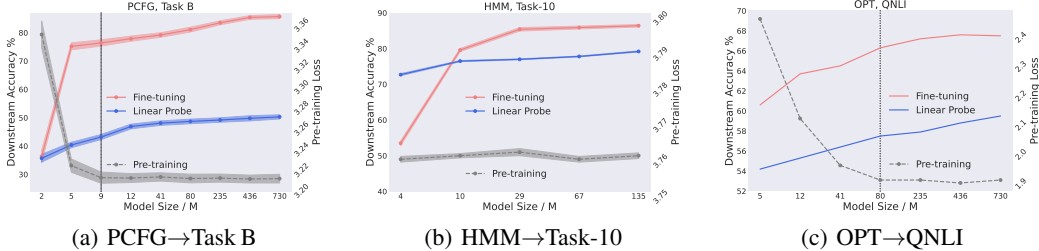

| (a) PCFG→Task B | (b) HMM→Task-10 | (c) OPT→QNLI |

Figure 2: Larger models perform better downstream than smaller models, even with almost the same pre-training loss. (a) Pre-train on the PCFG-generated dataset and evaluate on task B. (b) Pre-train on the HMM-generated dataset and evaluate on task-10. (c) Pre-train on the OPT-generated dataset and evaluate on QNLI. See Section 2 and Section A for details.

    quantities associated with the sentence such as sentiment and syntax. Thus we design three downstream tasks A, B, and C to classify non-terminal symbols at different positions of the parse trees.

2) *HMM-generated dataset.* HMM samples the hidden variables from the transition probabilities and the tokens from the emission probabilities. (Wei et al., 2021; Xie et al., 2021) also analyze the properties of pre-trained language models with HMMs. We generate the transition and emission probabilities as random block-diagonal stochastic matrices. The downstream task is to classify the hidden variable in the sentence. We use task-$k$ to refer to classifying the $k$-th hidden variable.

3) *OPT-generated dataset.* We also introduce a more realistic pre-training dataset generated by the OPT models (Zhang et al., 2022a). Starting from the `bos` token, we sample each token from the conditional probability output by the OPT model. For computational feasibility we only allow to generate the top-2000 most frequent tokens in the OPT vocabulary. We use QNLI and SST-2 from GLUE (Wang et al., 2018) as downstream tasks.

Note that the true conditional probability can be computed efficiently for the three datasets given the knowledge of the generated models. For PCFG and HMM-generated datasets, we can compute the true conditional probability with the inside algorithm (Lari & Young, 1990) and the Viterbi algorithm (Forney, 1973), respectively. For the OPT-generated dataset, we can calculate the MLM conditional probability from the joint probability, and the joint probability can be decomposed into the autoregressive conditional probability of the OPT model.

**Models and algorithms.** For PCFG and OPT generated datasets, we use transformers (Vaswani et al., 2017) following the implementation of BERT (Devlin et al., 2018). We use different model sizes ranging from 2M to 730M. For the HMM generated dataset, we use LSTM (Hochreiter & Schmidhuber, 1997) from 10M to 135M. In pre-training, all the models are pre-trained with AdamW following the protocol of Izsak et al. (2021). We use batch size 4096, and train each model for more than 30K steps until the pre-training loss converges. For comparison, we also consider other training algorithms. The first adversarial algorithm is inspired by Liu et al. (2020); Raghu et al. (2021), where the models are pre-trained with an additional meta-learning objective which messes up downstream performance. The second algorithm is manually setting the weight of the model to represent a lookup table which memorizes all the masked sentences and the corresponding true conditional probability. When the transformer is sufficiently large, the lookup table can be encoded as shown in Yun et al. (2019).

## 2.3 RESULTS

We compare the downstream performance of models with the same pre-training loss in the following situations: (1) training for different number of steps after the pre-training loss converges, (2) using different model sizes, and (3) training with normal training algorithms vs. adversarial training algorithms.

In Figure 1, we plot the validation pre-training loss and the downstream performance of different models checkpoints along pre-training. After the pre-training loss converges, although the pre-training loss does not improve, the downstream accuracy continues increasing.

Even with the same pre-training loss, larger models are better than smaller models. In Figure 2, we plot the pre-training loss and the downstream performance of models with different sizes. As we increase the model size, the pre-training loss approaches the entropy of the true conditional probability, which is 3.196, 3.758, and 1.865 for PCFG, HMM, and OPT respectively. For PCFG and OPT-generated datasets, we use the vertical dashed line to indicate the place where the pre-training loss saturates as

we scale up the model. For the much simpler HMM, the smallest 4M model can fit pre-training close to the entropy of the true conditional probability. With the same pre-training loss, scaling up the models improves linear probe performance by $6.9\%$, $4.5\%$, and $2.0\%$, on PCFG, HMM, and OPT generated data, respectively. See Section A for results on other downstream tasks.

Naturally trained transformers are better than adversarially trained ones. In Table 1, we evaluate the 235M transformers on PCFG tasks A and B with different pre-training algorithms. Although the adversarially trained transformer has almost the same pre-training loss as the normally trained 235M trans-

Table 1: Different pre-training algorithms on PCFG.

| Method | Pre-training | Task A % | Task B % |
|---|---|---|---|
| AdamW | 3.204 | 89.9 ±0.3 | 49.2±0.8 |
| Adversarial | 3.206 | 83.1 ±0.6 | 42.3±1.5 |
| Lookup table | 3.196 | 71.2 | 39.7 |

former, it is more than $6\%$ worse than the normal 235M model, and even worse than a normal 9M model on the downstream task B. The lookup table has perfect pre-training loss, but it performs worse than all normally trained transformers in Figure 2(a) on task B. Note that this is different from the label-orthogonal training in Saunshi et al. (2022). They find out models with the same pre-training loss and different downstream performance by subtracting the mean of the representations, essentially changing the architecture, while our experiment compares models with the same architecture.

The experiments above indicate that for models with the same architecture family and the same pre-training loss, the choice of training algorithms, model sizes, and the number of steps that the optimizer works can affect the downstream performance. This indicates the existence of *implicit bias* of the training algorithms toward choosing more transferable parameter configurations among those with the same pre-training loss and architecture.

## 3 IMPLICIT BIAS LEADS TO FLAT SOLUTIONS IN LANGUAGE MODELING

As discussed in the introduction, the difference between the role of implicit bias in supervised learning and language modeling is conceptual, because the gap between empirical and population self-supervised loss is small and thus implicit bias is not needed for bridging this gap. Instead, the implicit bias benefits the performance on downstream tasks by picking networks that are more adaptable to those tasks.

Fortunately, the mathematical tools developed for supervised learning can be adapted to language modeling, which even allows cleaner results by removing some artificial assumptions like adding label noise. Concretely, we show that mini-batch SGD can find models in the flatter areas of pre-training loss landscape. The flatness is measured by the trace of Hessian of the pre-training loss $\text{Tr}[\nabla^2 L(\theta)]$. See Figure 4 (Left) for an illustration of the implicit bias.

We analyze SGD on the population cross-entropy loss $L(\theta) = \mathbb{E}_{x,t}[-\log([f_\theta(x_{-t})]_{x_t})]$ with freshly sampled data at every iteration, because, as argued, the difference between empirical and population pre-training loss is not our focus. For simplicity, we present the results for batch size $=1$, though they can be generalized to arbitrary batch size (see discussion below Theorem 3.3). Let $\eta$ be the learning rate and let $\theta_k^\eta$ denote the parameter at step $k$. We drop the superscript $\eta$ when there is no ambiguity. We will show that the implicit bias kicks in when SGD reaches a global minimizer—it drives the iterate towards flatter global minimizers. For simplicity of demonstration, we analyze the process starting from a global minimizer $\bar{\theta}$, *i.e.*, we assume that $\theta_0^\eta = \bar{\theta}$ (for all $\eta$). At each iteration $k$, we get a fresh sample $(x,t)$, where $x$ is a sentence and $t$ is the position of the masked token, and update the parameter $\theta$ by $\theta_{k+1} = \theta_k - \eta \nabla_\theta \ell(f_{\theta_k}(x_{-t}), x_t)$. We assume the network is sufficiently expressive such that there are many fundamentally different global minimizers of the pre-training loss $L$. As a (non-trivial) regularity condition, following prior works (Fehrman et al., 2020; Li et al., 2021; Arora et al., 2022), we also assume that the minimizers of the loss function $L$ are connected and form a smooth manifold.

**Assumption 3.1.** *Assume that the loss $L$ is a $\mathcal{C}^3$-smooth function, and that the set of global minimizers, $\Gamma$, is a $(d-M)$-dimensional $\mathcal{C}^2$-submanifold of $\mathbb{R}^d$ for some integer $1 \le M \le d$, where for all $\theta \in \Gamma$, $\text{rank}(\nabla^2 L(\theta)) = M$.*

A key observation for language model is that even if the model reaches the saturation regime, that is, the model reaches a point on the manifold $\Gamma$ of the minimizers, the optimization process still has non-vanishing gradient noise, because the cross-entropy loss is typically *non-zero* at the global minimizers and thus the stochastic gradient variance is also non-zero.[2] Therefore, the dynamics of

---

[2]This is in contrast with typical supervised setting where the empirical 0-1 loss and cross-entropy loss can both achieve zero and consequently the mini-batch noise vanishes. Such a difference enables us to prove cleaner results (without the label noise) than in the supervised setting (Damian et al., 2021; Li et al., 2021).

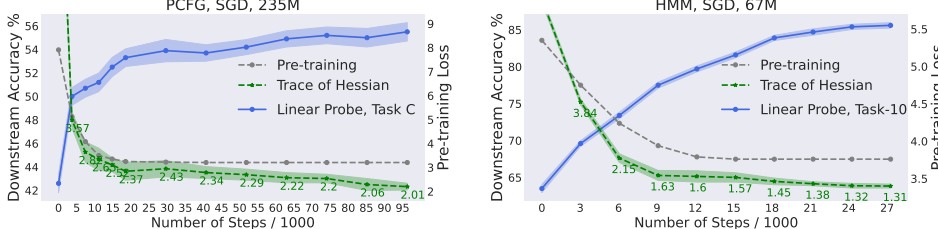

Figure 3: The trace of Hessian correlates with downstream performance for model checkpoints with different number of steps after the pre-training loss converges. Left: A model with 235M parameters pre-trained on the PCFG-generated dataset, and evaluated on task C. Right: A model with 67M parameters model pre-trained on the HMM-generated dataset, and evaluated on task-10.

SGD do not completely stop; instead, the iterate oscillates around the manifold $\Gamma$. It turns out that this oscillation in turn encourages the parameter to move in a certain direction along the manifold, determined by the covariance structure of the stochastic gradient. The following lemma shows that the covariance of stochastic gradient for language models in the saturating regime has a favorable property, *i.e.*, it is equal to the Hessian of pre-training loss.

**Lemma 3.2** (Bartlett identity). *For any $\theta \in \Gamma$, $\Sigma(\theta) = \nabla^2 L(\theta)$, where $\Sigma(\theta)$ is the covariance of the stochastic gradient at $\theta$, that is, $\Sigma(\theta) = \mathbb{E}_{t,x} \left[ \nabla_\theta \log[f_\theta(x_{-t})]_{x_t} (\nabla_\theta \log[f_\theta(x_{-t})]_{x_t})^\top \right] - \nabla L(\theta)^\top \nabla L(\theta)$.*

Though we give a proof of the lemma in Appendix C for completeness, the formula holds for the MLE loss of any well-specified probabilistic models at a global minimizer, and both the gradient covariance and the Hessian equals to the Fisher information matrix.

With Lemma 3.2, we can invoke Corollary 5.2 of Li et al. (2021) to derive the following theorem which says that SGD will locally decrease the trace of Hessian along the solution of ordinary differential equation (1) defined below.

$$\mathrm{d}\hat{\theta}(t) = -\frac{1}{4}\nabla_\Gamma \mathrm{Tr}[\nabla^2 L(\hat{\theta}(t))]\mathrm{d}t, \quad \hat{\theta}(0) = \overline{\theta} \tag{1}$$

where $\nabla_\Gamma = P_\Gamma^\perp \nabla$ is the Riemannian gradient on manifold $\Gamma$, or just the ordinary gradient projected back to the tangent space of $\Gamma$ at $\theta$. In other words, the ODE (1) is essentially a projected gradient descent algorithm with loss function $\mathrm{Tr}[\nabla^2 L(\theta)]$, the constraint set $\Gamma$, and infinitesimal learning rate. We show that SGD effectively minimizes the trace of the Hessian $\mathrm{Tr}[\nabla^2 L(\theta)]$ with the constraint set $\Gamma$ similarly to ODE in (1).

**Theorem 3.3.** *Suppose the loss function $L$ and the manifold of global minimizers $\Gamma$ satisfy Assumption 3.1. For any $K > 0$ such that ODE (1) has a solution $\{\hat{\theta}(t)\}_{t=0}^K$, it holds that $\theta_{K/\eta^2}^\eta$ converges in distribution to $\hat{\theta}(K)$ as $\eta \to 0$.*

Finally, we note that the above result can be extended to an arbitrary batch size $B$. The covariance of stochastic gradient at $\theta$ with batch size, denoted by $\Sigma_B(\theta)$, satisfies that $\Sigma_B(\theta) = \frac{1}{B}\Sigma(\theta)$. Therefore $\Sigma_B(\theta) = \frac{1}{B}\nabla^2 L(\theta)$ and we can again invoke Corollary 5.2 of Li et al. (2021) to derive the same result as in Theorem 3.3 but with the coefficient $\frac{1}{4}$ in equation (1) replaced by $\frac{1}{4B}$.

## 4 FLATTER MODELS HAVE BETTER DOWNSTREAM PERFORMANCE

In this section, we demonstrate with experiments that the flatness is well correlated with downstream performance in the setting introduced in Section 2.

**Evaluation of flatness.** As in Theorem 3.3, we measure the flatness of different models in Section 2 by the trace of Hessian of the pre-training loss (smaller trace of Hessian indicates flatter minima.) Note that when the model approaches the saturation regime, the trace of Hessian is approximately the second order derivative times the square of the norm of the Jacobian, which is a high-dimensional matrix. For computational feasibility, we adopt a technique inspired by Wei et al. (2020) to unbiasedly estimate the trace of Hessian with random samples. Details are provided in Section B.

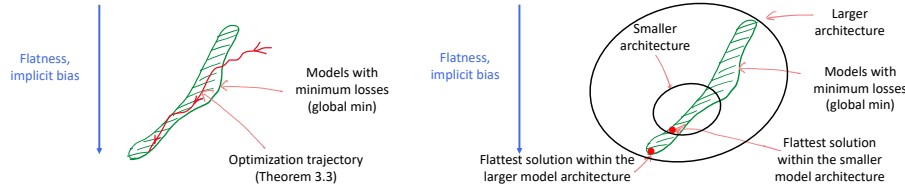

Figure 4: Left: The role of implicit bias. After the pre-training loss converges, the implicit bias drives the model toward flat minima, as predicted by Theorem 3.3. Right: The interaction between model size and implicit bias. The implicit bias drives the model toward flat minima on both larger models and smaller models. The smaller model architecture can be viewed as a subset of the larger model architecture. Therefore, larger models can achieve flatter minima than smaller models.

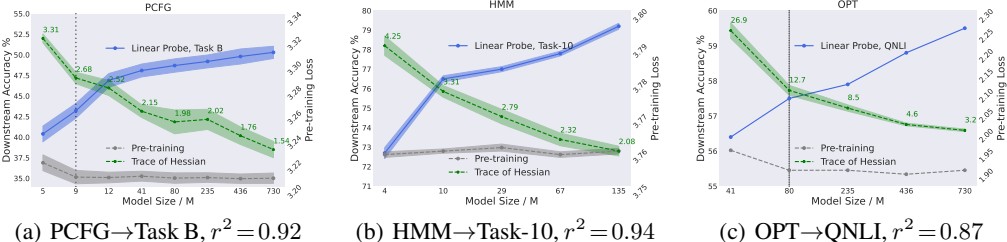

(a) PCFG→Task B, $r^2 = 0.92$       (b) HMM→Task-10, $r^2 = 0.94$       (c) OPT→QNLI, $r^2 = 0.87$

Figure 5: The trace of Hessian correlates with downstream performance for models with different sizes and almost the same pre-training loss. On datasets generated by PCFG, HMM and OPT, we obeserve 0.92, 0.94, and 0.87 coefficient of determination with linear regression.

**Results.** In Figure 3, we compare the downstream accuracy and the trace of Hessian of different checkpoints obtained at different times during pre-training. On the PCFG and HMM datasets, the trace of Hessian demonstrates a clear decreasing trend after the validation pre-training loss converges, following the

Table 2: A 235M transformer pre-trained with different algorithms evaluated on PCFG Task C.

| Method | Pre-training | Acc % | Tr(H) |
|---|---|---|---|
| AdamW | 3.20 | 55.7±0.6 | 2.02±0.16 |
| Adversarial | 3.20 | 50.2±1.0 | 6.45±0.41 |

prediction of Theorem 3.3. Furthermore, as the trace of Hessian decreases, the downstream performance improves by 1.6% and 4.0% on the PCFG and HMM datasets, respectively.

We compare the trace of Hessian of the models pre-trained with adversarial algorithm and standard AdamW in Table 2. The trace of Hessian of the adversarially pre-trained model is 3 times larger than the normally pre-trained model, corresponding to a drop of 5.5% in downstream performance.

In Figure 5, we compare the downstream accuracy and the trace of Hessian of models with different sizes. On the dataset generated by a PCFG, the pre-training loss is almost the same for models larger than 9M. As we increase the model size, the trace of Hessian of the pre-training loss decreases from 2.68 to 1.54, correlating with the increase of linear probe accuracy from 43.2% to 50.3%. On the OPT and HMM-generated datasets, we can also observe an increase in linear probe accuracy with a sharp decrease in the trace of Hessian, as we increase the model size.

**Interaction between implicit bias and model size.** Intuitively, the implicit bias drives the model toward flat minima on both larger models and smaller models. The smaller transformer architecture is a subset of the larger transformer architecture family (as justified in Section B). Thus the flattest minimum found within a larger transformer is flatter than the flattest minimum found within a smaller transformer, and performs better downstream. (See Figure 4 (Right).)

## 5    FLATNESS REGULARIZATION PROVABLY IDENTIFIES TRANSFERABLE MODELS ON SYNTHETIC LANGUAGE

Toward formally proving the connection between the flatness regularization (introduced by the stochastic gradient as argued in Section 3) and the downstream performance, we consider a setting with synthetic Dyck language. The simplicity of the data allows us to sharply analyze the internal working of a single-layer transformer (with an attention layer and an MLP layer) for masked language modeling. We show that multiple parameter configurations can predict the conditional probability well, including one ideal model that learns the correct representations capturing the intrinsic structure of the sentence,

Figure 6: **The synthetic language setting.** Left: An example of input encodings with sentence length $T=6$. Right: Illustration of the two solutions. The softmax attention can sum the token encodings into $z$. Solution (1) contains two features transferable to the downstream task. The neurons in solution (2) are sampled from Gaussian distribution, and not related to the downstream task. Both solutions can output the correct prediction for MLM pre-training, but solution (1) has much smaller trace of Hessian.

and many "cheating" models that essentially memorize the conditional probability using random features. We will prove that the flattest model is the desired model that transfers to downstream tasks.

**Pre-training Distribution.** Consider a variant of the Dyck language (Nivat, 1970) consisting of matching brackets. The vocabulary of the language has two brackets $\langle$ and $\rangle$. Each sentence is composed of a sequence of tokens such that the total numbers of $\langle$'s and $\rangle$'s are equal. To sample from the pre-training distribution $P$, we first draw a sentence uniformly over all valid sentences with even length $T$. Then, we randomly select one position in $[T]$ and replace the bracket with a mask token.

**Downstream Task.** The most intrinsic property about the synthetic language is the difference in the number of left and right brackets in a prefix, and thus we use it as the downstream task. Concretely, for any sequence $x$ in $\{\langle,\rangle\}^*$ of length $T$, let $g^*(x)$ count the number of mismatches in $x$:

$$g^*(x) \triangleq \# \text{ of } \rangle\text{'s in } x - \# \text{ of } \langle\text{'s in } x \tag{2}$$

Thus, the sentence $x$ is a valid string in the language if and only if $g^*(x)=0$. For MLM, the masked token can also be recovered from $g^*(x)$: $g^*(x)=1$ if the masked token is $\langle$, and $g^*(x)=-1$ if the masked token is $\rangle$. To evaluate if the model learns the structure, we consider a downstream distribution of sentences which do not necessarily belong to the the language. Each token is sampled from $\{\langle,\rangle\}$ uniformly, randomly, and independently.

**Encoding of the Inputs.** With a slight abuse of notation, we also denote by $x_t$ the encoding of the $t$-th token. We encode the input as a one-hot vector in dimension $d = 2T$, where the index of the nonzero element encodes the position and the sign encodes the bracket. Concretely, let $e_t \in \mathbb{R}^d$ be the natural basis vector where the $t$-th entry is 1. Let $x_t = e_t$ if the $t$-th token is $\langle$ and $x_t = -e_t$ otherwise. If the position $t$ is a mask, we set $x_t$ to $v$, where $v \sim \text{Unif}(\{\pm e_{t+T}\})$. Examples of the encodings with $T=6$ are provided in Figure 6 (Left). Note that the target function can be expressed as $g^\star(x) = -\langle \mathbf{1}_T, [\sum_{t=1}^T x_t]_{1:T} \rangle$ with this input encoding, where $\mathbf{1}_T$ is the all one vector in $\mathbb{R}^T$ and $[a]_{1:T}$ refers to the first $T$ coordinates in $a$.

**Models and Algorithms.** Suppose $Q,K \in \mathbb{R}^{k \times d}$ are the query and key matrices, $V \in \mathbb{R}^{m \times d}$ is the value matrix and $u \in \mathbb{R}^m$ is the parameter of the output layer. Let $\psi = (Q,K,V)$. A single-layer transformer is composed of an attention layer and an MLP layer. $[\text{Attn}_{\psi,u}(x)]_t = \frac{1}{m} u^\top \sigma(\sum_{j=1}^T a_{t,j} V x_j)$, where the attention score $a_{t,1:T} = \text{softmax}(\langle Q x_t, K x_t \rangle, \cdots \langle Q x_t, K x_T \rangle)$. $\sigma(x) = \max\{x,0\}$ is the relu activation. We use the output of the first token, $f_{\psi,u}(x) = [\text{Attn}_{\psi,u}(x)]_1$.

We use the squared loss for both MLM and downstream adaptation. The loss function of MLM is $L(\psi,u)$. In downstream adaptation, we have a finite dataset $\{x^{(i)}\}_{i=1}^n$ sampled i.i.d. from $P_{\text{ds}}$. The training loss with $n$ data is $\widehat{L}^{P_{\text{ds}}}(\psi,u)$, and the population loss for the downstream task is $L^{P_{\text{ds}}}(\psi,u)$.

**Main Intuitions.** We are interested in two kinds of parameter configurations both with good pre-training loss: (1) learning the natural and transferable features $\mathbf{1}_T$ and (2) fitting the pre-training task by memorizing the masked sentences. We construct the two solutions as follows. For solution (1), first note that the softmax attention layer can take the average of all the token encodings $[x_t]_{t=1}^T$ in a sentence. Let us denote the sum by $z \in \mathbb{R}^d$, $z = \sum_{t=1}^T x_t$. Note that the first $T$ coordinates in $z$ are $\pm 1$ indicating the bracket type and the last $T$ coordinates indicate the position of the mask (See Figure 6 (Right)). On top of $z$, two neurons can predict the masked token in MLM perfectly. Consider the two neurons $V_1 = [\mathbf{1}_T; \mathbf{0}_T]$, $V_2 = [-\mathbf{1}_T; \mathbf{0}_T]$. Then $g^*(x) = \sigma(V_2^\top z) - \sigma(V_1^\top z)$, which is the transferable solution. For solution (2), we set the entries in $V$ to i.i.d. samples from $\mathcal{N}(0,T)$. If $m$ is sufficiently large, we can find the coefficient $u$ to express $g^*(x)$ with random Gaussian features, i.e. $g^*(x) = u^\top \sigma(Vz)$.

We observe that the trace of Hessian of configuration (1) is smaller than configuration (2), due to a main difference between them–the cancellation between activated neurons. In configuration (1), for every possible input, only one of the neurons $\sigma(V_1^\top z)$ and $\sigma(V_2^\top z)$ is activated. In contrast, in configuration (2), many neurons can be activated at the same time. Among them, the output coefficient $u_i$'s contain both positive and negative values, leading to cancellation between activated neurons. In Lemma F.1, we link the trace of the Hessian with the cancellation between neurons. Indeed, we show that the mimimum of trace of the Hessian can be achieved only if there is no such cancellation. Therefore solution (1) is also the minimizer of the trace of Hessian. The intuitions are formalized in Theorem 5.1.

Consider minimizing the trace of Hessian among all the solutions to the MLM pre-training task: $\text{minimize}_{\psi,u} \text{Tr}[\nabla_\psi^2 L(\psi,u)] + \text{Tr}[\nabla_u^2 L(\psi,u)]$, subject to $L(\psi,u) = 0$.

**Theorem 5.1.** *Suppose $m \geq 2$ and $T \geq 6$. The flattest solution $\hat{\psi}, \hat{u}$ are defined as the solution of the optimization problem above. $\tilde{u}$ is the minimizer of downstream training loss on top of $\hat{\psi}$, $\tilde{u} \in \text{argmin}_u \|u\|_2$ subject to $\widehat{L}^{P_{ds}}(\hat{\psi}, u) = 0$. Then with probability at least $1 - 2^{-n}$, $L^{P_{ds}}(\hat{\psi}, \tilde{u}) = 0$.*

## 6 RELATED WORK

**Language modeling and downstream adaptation.** Large language modeling has revolutionized the NLP field. Starting from Devlin et al. (2018), a line of works improve the downstream performance on a wide range of tasks with increasing model size and data amount (Yang et al., 2019; Radford et al., 2019; Raffel et al., 2020). LLMs even exhibit unexpected emergent behaviors, such as in-context learning (Xie et al., 2021; Min et al., 2022), step-by-step reasoning (Wei et al., 2022), and zero-shot learning (Brown et al., 2020). Kaplan et al. (2020); Hernandez et al. (2021) study the behavior of language models with increasing size, and find out that the pre-training loss is typically correlated with downstream performance as model size increases. In practice, the pre-training loss is used as an evaluation metric for language models. A notable example is the efficient transformer line of works, which benchmark the pre-training loss given the same computation constraint (Dai et al., 2020; Wang et al., 2020; Choromanski et al., 2020; Liu et al., 2021).

**Understanding the success of language modeling.** Empirical works on understanding MLM find out that the representations of language models encode rich semantic and syntactic information (Peters et al., 2018; Htut et al., 2019; Hewitt & Manning, 2019; Mamou et al., 2020). Theoretical works show that fitting the MLM conditional probability is a sufficient condition for good performance on downstream tasks. Zhang & Hashimoto (2021) show MLM representations recover latent variables in graphical models. Wei et al. (2021) show linear probe on top of MLM models solves downstream tasks on datasets generated by HMMs. Recently, Saunshi et al. (2022) show that models with the same pre-training loss but different architectures can have different downstream performance. Tay et al. (2021) find out that a narrow but deep transformer is better than a wide but shallow transformer with the same pre-training loss. Zhang et al. (2022b) demonstrate that Albert (Lan et al., 2019) generalizes better to OOD tasks than Bert on a synthetic reasoning task. These works indicate that the architecture is an important factor for good downstream performance beyond pre-training loss. This paper discovers the role of implicit bias in language modeling, which happens with models in the same architecture.

**Implicit bias in supervised learning.** The training algorithm chooses solutions with certain properties, and usually leads to better generalization (Gunasekar et al., 2018; Soudry et al., 2018; Li et al., 2017; Ji & Telgarsky, 2018; Arora et al., 2019; Lyu & Li, 2019; Li et al., 2020; Woodworth et al., 2020; HaoChen et al., 2020). Recently, Blanc et al. (2019); Damian et al. (2021); Li et al. (2021) demonstrate label noise SGD biases the models toward flatter minima. However, the setting of implicit bias in supervised learning is different from language modeling. In language modeling, we have access to gigantic corpus, and cannot interpolate the pre-training dataset. Moreover, we care about the adaptability of the solution on downstream tasks instead of generalization in distribution.

## 7 CONCLUSION

We study the relationship between pre-training loss and downstream performance on language models. We discover that implicit bias matters beyond pre-training loss, and explore the mechanism of implicit bias in language modeling. Our experiments focus on simplified datasets due to constraint of computational resources. We hope that the phenomenon can predict the implicit bias on more complex datasets as the community scale the models to even larger. Theoretically we provide cases where the flatness regularization can decide the performance on downstream performance. We wish this motivates future works on the relationship between implicit bias and the internal working of the models.

## REPRODUCIBILITY STATEMENT

To ensure reproducibility, we describe the implementation details of the algorithms and the construction of the datasets in Section A and Section B. The code of the experiments is provided in the supplementary material. We provide the proof in Section C and Section F.

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

## A DETAILS IN SECTION 2

### A.1 GENERATING SIMPLIFIED DATASETS

**PCFG-generated dataset.** We consider a PCFG with vocabulary size 200. The state space is $S$, and $|S|=50$. All the production rules have two symbols on the right side. The sentence length is limited to 32, which means the depth of the parse tree is limited to 6. We generate a total of $2\times10^7$ sentences, which is $3.4\times10^8$ tokens. The downstream tasks are classifying the non-terminal symbols in the parse tree of the PCFG (50-way classification). The label is defined as $y=\mathrm{argmax}_{s\in S}\mathrm{Pr}(s\,|\,x_1,x_2,...,x_{1+L})$. Tasks A, B and C are defined on the symbols corresponding to span length $L=32,16$ and 8, respectively. Each of the downstream task contains 0.1M examples. Examples of the generated trees are provided in Figure 7.

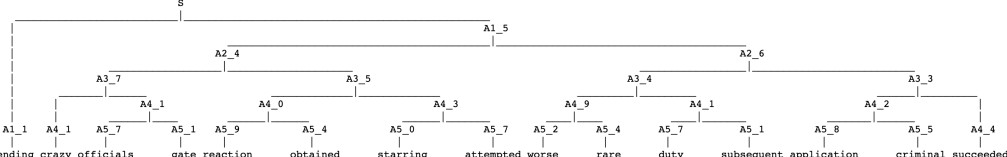

Figure 7: An example of the genearated PCFG sentence.

**HMM-generated dataset.** We consider an HMM with vocabulary size 200 and state space size 100. The sentence length is restricted to 16. We generate a total of $1\times10^7$ sentences, which is $1.6\times10^7$ tokens. The downstream task is to classify the latent variable in the HMM generative model. We consider task-6 and task-10, which classify the 6-th and 10-th hidden variables respectively. Each of the downstream task contains 0.1M examples.

**OPT-generated dataset.** We use the 125M OPT model to generate the training dataset. To simplify the dataset, we further process the logit of OPT to select only from the top-2000 tokens in the vocabulary. Starting from the bos token, we sample every token of the sentence from the predicted autoregressive LM probability. The sentence length is restricted to 24. We generate a total of $2\times10^8$ sentences, which is $3.2\times10^9$ tokens. Examples of the generated text are provided in Figure 8.

```
I really don't either, so why do you feel it wouldn't be great when you can
I want the person in the photo to tell me there are children under 6 years old in the
```

Figure 8: An example of the genearated OPT sentence.

### A.2 COMPUTE THE TRUE CONDITIONAL PROBABILITIES

We can compute the true MLM conditional probability $\mathrm{Pr}(x_t\,|\,x_{-t})$ from the joint probability $\mathrm{Pr}(x_t,x_{-t})$ with one mask per token,

$$\mathrm{Pr}(x_t=c\,|\,x_{-t})=\frac{\mathrm{Pr}(x_t=c,x_{-t})}{\sum_{c\in W}\mathrm{Pr}(x_t=c,x_{-t})}.$$

Since we already know the generative model, we can compute the joint probability efficiently. For PCFG, we can compute the joint probability with the inside algorithm, which decomposes the joint probability into lower layers in the parse tree. For HMM, we can compute the joint probability with the Viterbi algorithm. For OPT, we have $\mathrm{Pr}(x_1,...,x_T)=\mathrm{Pr}(x_1)\prod_{t=1}^{T-1}\mathrm{Pr}(x_{t+1}\,|\,x_1,...,x_t)$.

### A.3 MODELS

We use transformers on PCFG and OPT-generated datasets. We use learning rate 1e-3 and warmup proportion 0.06. All the models are trained based on the implementation of Izsak et al. (2021). We list the sizes of the transformers in Table 3. d-model is the size of the hidden layers. d-inter is the size of the

Table 3: Shape of the transformers.

| d-model | n-nead | #layers | d-inter |
|---------|--------|---------|---------|
| 64 | 1 | 1 | 256 |
| 128 | 2 | 4 | 512 |
| 192 | 3 | 5 | 768 |
| 256 | 4 | 6 | 1024 |
| 512 | 8 | 8 | 2048 |
| 768 | 12 | 12 | 3072 |
| 1024 | 16 | 16 | 4096 |
| 1280 | 20 | 20 | 5120 |
| 1536 | 24 | 24 | 6144 |

intermediate layers in MLP. n-head is the number of heads per layer. #layers is the number of layers. For LSTMs, we use the implementation of PyTorch. We consider d-hidden in [128,256,512,768,1024], and #layers in [4,6,8,12,16].

### A.4 ALGORITHMS

**The lookup table.** To evaluate the downstream performance of the lookup table, we first create the lookup table with the data of the downstream task. With the method mentioned in Section A.2, we can generate the true conditional probability of each token and use it as the contextual embeddings.

**The adversarial algorithm.** The adversarial algorithm we use to mess up the downstream performance is maximizing a meta-learning objective in pre-training. Suppose the linear head of the downstream task is $g_\phi$ and the feature representation is $h_\psi$. The meta-learning algorithm first trains the head $g_\phi$ to minimize the training loss of the downstream task, and then update $h_\psi$ to maximize the validation loss on the downstream tasks. Concretely, we randomly sample two disjoint subsets $D_1$ and $D_2$ from the downstream training dataset $D$. We train $g_\phi$ to minimize the loss of downstream tasks on $D_1$, $\hat{\phi}(\psi) \in \text{argmin} \frac{1}{|D_1|} \sum_{(x,y) \in D_1} \ell(g_\phi(h_\psi(x)), y)$. Then we train $h_\psi$ to maximize the validation loss on $D_2$ during pre-training, $\text{minimize}_\psi L(\psi) - \lambda \frac{1}{|D_2|} \sum_{(x,y) \in D_2} \ell(g_{\hat{\phi}(\psi)}(h_\psi(x)), y)$. The optimization can be efficiently carried out with closed form solution of $\phi$ as shown in Liu et al. (2020).

**Fine-tuning.** Following the standard protocol of Devlin et al. (2018), we use the contextual embeddings of the CLS token for fine-tuning. We use AdamW with learning rate 1e-4. We perform 200 warmup steps and train on the downstream tasks for 10 epochs.

**Linear probe.** Since the CLS token is not trained in pre-training, we concatenate embeddings of all the tokens in the sentence as the representations. We use AdamW with learning rate 1e-3 to train the linear head. We train on the downstream tasks for 100 epochs. Note that to make the capacity of the linear probe itself controlled, we adopt a random Gaussian projection to dimension 512 on the concatenation of the embeddings.

We report the standard deviation of linear probe and fine-tuning from 5 random seeds.

**Evaluation of pre-training loss.** Since we have access calculate the true conditional probability, we can calculate the cross entropy loss as the sum of the entropy of the true conditional probability and the KL divergence between the predicted and true conditional probabilities. This is more accurate than evaluating on the validation datasets in the standard ways. We report the number of pre-training loss with $10^6$ sentences, and calculate the standard deviation on 5 subsets, each of which has size $2 \times 10^5$.

### A.5 RESULTS ON OTHER DOWNSTREAM TASKS.

We also provide results on other downstream tasks in this subsection. On PCFG Task A, OPT SST-2 and the Task-6 of HMM, we can also observe the increase in downstream performance as we scale up the models in the saturation regime.

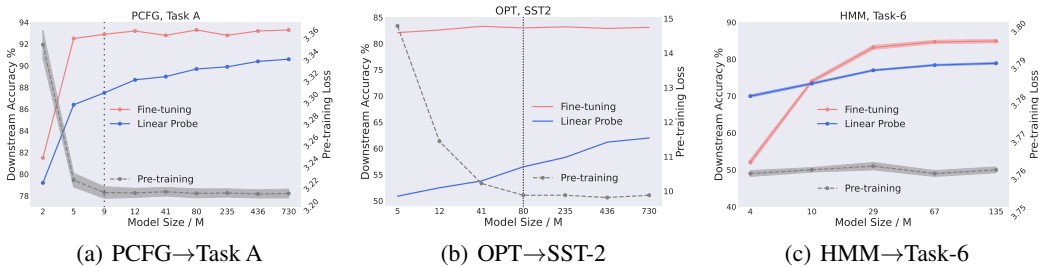

Figure 9: The downstream accuracy continues to rise as we increase the model size, although the pre-training loss remains unchanged.

## B   DETAILS IN SECTION 4

**Unbiased estimate of the trace of Hessian.**   Evaluating the trace of Hessian requires the norm of the Jacobian $\nabla_\theta \log[f_\theta(x_{-t})]$. Since the output dimension $c$ and the number of parameters are all very large, computing the Jacobian $\nabla_\theta \log[f_\theta(x_{-t})]$ will be very inefficient. Instead, we can estimate the trace of Hessian unbiasedly with random samples as follows. Suppose $f_\theta(x_{-t})$ is the predicted probability of the conditional probability. In the saturation regime, as $f_\theta(x_{-t})$ approaches the true conditional probability, the Hessian of the pre-training loss w.r.t. the parameters can be expressed as

$$\nabla^2 L(\theta) = \mathbb{E}_{t,x_{-t}} \mathbb{E}_{x_t|t,x_{-t}} \left[ \nabla_\theta \log[f_\theta(x_{-t})]_{x_t} (\nabla_\theta \log[f_\theta(x_{-t})]_{x_t})^\top \right].$$

Therefore we have

$$\mathrm{Tr}(\nabla^2 L(\theta)) = \mathbb{E}_{t,x_{-t}} \mathbb{E}_{x_t|t,x_{-t}} \|\nabla_\theta \log[f_\theta(x_{-t})]_{x_t}\|_2^2.$$

To approximate this expectation, we can first sample $t, x_{-t}$ from the language, then draw i.i.d. samples $x_t$ from $(f_\theta(x_{-t}))$, and use the average as the unbiased estimate. For all experiments, we sample 10000 $x_{-t}$ and sample 50 $x_t$ for each $x_{-t}$.

**Details in Figure 3.**   To verify Theorem 3.3 that SGD biases the model towards flatter minima, we conduct MLM on PCFG and HMM-generated datasets with SGD. We set the proportion of warmup stage to 12% total number of steps, and fix the learning rate to 1e-3 after the warmup. We evaluate the downstream performance and the trace of Hessian of different checkpoints along pre-training. The standard deviation of trace of Hessian is calculated based on 5 times of sampling 50 examples as mentioned above. Apart from the PCFG task C and HMM task-10, we also provide results on PCFG tasks A, B and HMM task-6 in Figure 10.

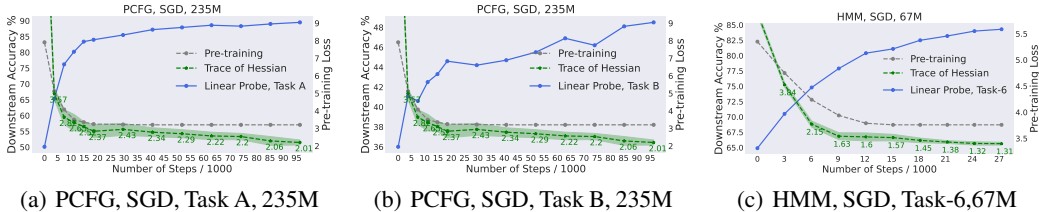

Figure 10: The trace of Hessian correlates with downstream performance for model checkpoints with different number of steps after the pre-training loss converges.

### B.1   EMBEDDING OF A SMALLER TRANSFORMER INTO A LARGER TRANSFORMER.

In this subsection, we show that a smaller transformer can be embed into a larger transformer without changing the functionality. We enable the embedding by considering two techniques (1) adding additional layers using residual connections without changing the functionality and (2) increasing feature dimension / adding more attention heads without change the functionality by duplicating the weights.

### B.1.1    THE BASE CASE WITH MLPS.

To gain some insights of how to increase the feature dimension without changing the functionality, we start with vanilla MLPs without layer-norm and residual connections He et al. (2016). Consider a multi-layer MLP $f_{W,a}(x)$. The weight matrices are $W = [W_0,...,W_{L-1}]$. The dimensionality is $W_l \in \mathbb{R}^{d_{l+1} \times d_l}$. The representations are defined recursively, $h_{l+1}(x) = \sigma(W_l h_l(x))$. We denote the input by $h_0(x) = x$ and the final output is defined as $f_{W,a}(x) = a^\top h_L(x)$. The activation $\sigma$ here is relu or leaky relu. We aim to embed $f_{W,a}(x)$ into $f_{\tilde{W},\tilde{a}}(x)$, where $\tilde{W} = [\tilde{W}_0,...,\tilde{W}_{L-1}]$ and $\tilde{W}_l \in \mathbb{R}^{2d_{l+1} \times 2d_l}$. We need to make sure $f_{W,a}(x) = f_{\tilde{W},\tilde{a}}(x)$ for all $x$.

For this case, we can set $\tilde{W}_l = 1/2 \begin{bmatrix} W_l & W_l \\ W_l & W_l \end{bmatrix}$ for $l \in [L-1]$, $\tilde{W}_0 = \frac{1}{\sqrt{2}} \begin{bmatrix} W \\ W \end{bmatrix}$, and $\tilde{a} = \frac{1}{\sqrt{2}} \begin{bmatrix} a \\ a \end{bmatrix}$. We can verify that $\tilde{h}_l(x) = \frac{1}{\sqrt{2}} \begin{bmatrix} h_l(x) \\ h_l(x) \end{bmatrix}$ inductively. Therefore we have

$$\begin{aligned} f_{\tilde{W},\tilde{a}}(x) &= \tilde{a}^\top \tilde{h}_L(x) \\ &= \frac{1}{\sqrt{2}} [a^\top, a^\top] \frac{1}{\sqrt{2}} \begin{bmatrix} h_L(x) \\ h_L(x) \end{bmatrix} \\ &= \frac{1}{2} f_{W,a}(x) + \frac{1}{2} f_{W,a}(x) \\ &= f_{W,a}(x). \end{aligned}$$

### B.1.2    REAL TRANSFORMERS.

Next we turn to transformers with residual connections and layer norm. We first use the same strategy as the MLP case to add additional feature dimension and attention heads by replicating the weights, and then show how to add new layers using residual connections. At a high level, replicating the weight maintains the mean and the variance calculated by the layernorm. Therefore the representations inside the transformer also get replicated, without changing the values in each of the replicated groups.

**Setup.** A transformer is composed of an input embedding $W_E$, $L$ blocks of self-attention, and an output layer. Transformers also contain layer-norm and residual connections. Suppose the input $x = [x_1,...,x_T]$, where $x_i \in \mathbb{R}^d$. Each block of the self-attention contains of an attention layer and an MLP layer, both equipped with residual connections and layer-norm. Let us denote by $[h_0(x)]_i = \text{LN}(W_E x_i)$ the input embeddings. Suppose the hidden size is $d_h$, i.e. $[h_l(x)]_i \in \mathbb{R}^{d_h}$. The attention layer is defined as $[v_l(x)]_i = \text{LN}([h_l(x)]_i + [\text{Attn}_l(h_l(x))]_i)$, and the MLP layer is defined as $[h_{l+1}(x)]_i = \text{LN}([v_l(x)]_i + U_l \sigma(W_l [v_l(x)]_i + b_l))$. The activation $\sigma$ is GeLU. The final output is $[f(x)]_i = W_E^\top [h_L(x)]_i$. Note that $W_E$ is both the input embedding and the weight of the output layer. They are tied in training.

The layer norm is on the feature dimension. $[\text{LN}(x_i)]_j = \gamma_j * \hat{x}_{ij} + \beta_j$. $\hat{x}_i$ is the normalized version of $x_i$ with zero mean and unit variance. $\gamma$ and $\beta$ are trainable.

The multi-head attention consists of $n_h$ self-attention heads. The definition of the multi-head attention is $\text{Attn}_l(h_l(x)) = [(A_{l1} h_l(x) V_{l1})^\top,...,(A_{ln_h} h_l(x) V_{ln_h})^\top]^\top O_l$. The output matrix $O_l \in \mathbb{R}^{d_h \times d_h}$. The attention heads composes of the attention score times the feature matrix times the value matrix. The attention score $A_{lk} \in \mathbb{R}^{d' \times d'}$ is computed with softmax dot product. For each $k \in [n_h]$, $A_{lk} = \text{softmax}(h_l(x) Q_{lk} K_{lk}^\top h_l(x)^\top)$.

Following the implementation of Devlin et al. (2018), the dimension of the attention head is always $d' = 64$, thus $d_h = 64 n_h$. The dimension of the intermediate layer in the MLP is set to $4d_h$, which means $U_l \in \mathbb{R}^{d_h \times 4d_h}$ and $W_l \in \mathbb{R}^{4d_h \times d_h}$.

We aim to embed the smaller transformer $f(x)$ into $\tilde{f}(x)$, where $\tilde{d}_h = 2d_h$, $\tilde{n}_h = 2n_h$, and $\tilde{L} = L + L'$.

**Increasing feature dimension with replication of the parameters.** Although the transformers have layer-norm and residual connections, we can still modify the strategy in the base case with MLPs slightly to increase the width of the model and the number of attention heads without changing the

functionality. Consider the following weight replication method. For $l \in [0,...,L-1]$,

$$\tilde{W}_E = \frac{1}{2}\begin{bmatrix} W_E \\ W_E \end{bmatrix}, \tilde{\gamma}_l = \begin{bmatrix} \gamma_l \\ \gamma_l \end{bmatrix}, \tilde{b}_l = \begin{bmatrix} b_l \\ b_l \end{bmatrix}, \tilde{\beta}_l = \begin{bmatrix} \beta_l \\ \beta_l \end{bmatrix},$$

$$\tilde{W}_l = \frac{1}{2}\begin{bmatrix} W_l & W_l \\ W_l & W_l \end{bmatrix}, \tilde{U}_l = \frac{1}{2}\begin{bmatrix} U_l & U_l \\ U_l & U_l \end{bmatrix}, \tilde{O}_l = \frac{1}{2}\begin{bmatrix} O_l & O_l \\ O_l & O_l \end{bmatrix},$$

$$\tilde{Q}_{lk} = \frac{1}{2}[Q_{lk}, Q_{lk}], \tilde{K}_{lk} = \frac{1}{2}[K_{lk}, K_{lk}], \tilde{V}_{lk} = \frac{1}{2}[V_{lk}, V_{lk}] \text{ for } k \in [1,...,n_h],$$

$$\tilde{Q}_{lk} = \frac{1}{2}[Q_{l(k-n_h)}, Q_{l(k-n_h)}], \tilde{K}_{lk} = \frac{1}{2}[K_{l(k-n_h)}, K_{l(k-n_h)}], \tilde{V}_{lk} = \frac{1}{2}[V_{l(k-n_h)}, V_{l(k-n_h)}] \text{ for } k \in [n_h+1,...,2n_h].$$

We observe that the intermediate layers of the transformers are also replicated for the first $L$ blocks, i.e. $\tilde{h}_l(x) = \begin{bmatrix} h_l(x) \\ h_l(x) \end{bmatrix}$ and $\tilde{v}_l(x) = \begin{bmatrix} v_l(x) \\ v_l(x) \end{bmatrix}$ for $l \in [1,...,L]$. First note that $[h_0(x)]_i = \text{LN}(W_E x_i)$. Since replicating the features will not change the mean and the variance, we have $\tilde{h}_0(x) = \begin{bmatrix} h_0(x) \\ h_0(x) \end{bmatrix}$. Then we can show that replicating the features will not change the attention scores as well. This makes $\tilde{v}_l(x) = \begin{bmatrix} v_l(x) \\ v_l(x) \end{bmatrix}$. Finally note that we can apply the base case of the MLP to reason about the MLP layer, and show $\tilde{h}_{l+1}(x) = \begin{bmatrix} h_{l+1}(x) \\ h_{l+1}(x) \end{bmatrix}$. Therefore we have shown $\tilde{h}_l(x) = \begin{bmatrix} h_l(x) \\ h_l(x) \end{bmatrix}$ inductively.

**Adding additional layers using residual connections.** We have demonstrated that $\tilde{h}_l(x) = \begin{bmatrix} h_l(x) \\ h_l(x) \end{bmatrix}$ for $l \in [0,...,L]$. Now let's consider the added $L'$ blocks on top of the small model. Since the transformer contains residual connections, we can add new blocks on top of a small model and fill in zeros to the added parameters. We will show that in this way, $\tilde{h}_l(x) = \tilde{h}_L(x)$, for any $l \in [L,...,L+L']$. This will indicate that $\tilde{h}_{L+L'}(x) = \tilde{h}_L(x) = \begin{bmatrix} h_l(x) \\ h_l(x) \end{bmatrix}$. Recall that $\tilde{W}_E = \frac{1}{2}\begin{bmatrix} W_E \\ W_E \end{bmatrix}$. This indicate that

$$\begin{aligned}
[\tilde{f}(x)]_i &= \tilde{W}_E^\top \tilde{h}_{L+L'}(x) \\
&= \frac{1}{2}[W_E^\top, W_E^\top]\begin{bmatrix} h_L(x) \\ h_L(x) \end{bmatrix} \\
&= \frac{1}{2}[f(x)]_i + \frac{1}{2}[f(x)]_i \\
&= [f(x)]_i,
\end{aligned}$$

which means we can add new layers on top of a small transformer without changing the functionality.

Now we show that $U_l = 0$ and $O_l = 0$ for $l \in [L,...L+L'-1]$ will make $\tilde{h}_l(x) = \tilde{h}_L(x)$, for any $l \in [L,...,L+L']$. This holds because $[v_l(x)]_i = \text{LN}([h_l(x)]_i + [\text{Attn}_l(h_l(x))]_i)$ and $[h_{l+1}(x)]_i = \text{LN}([v_l(x)]_i + U_l\sigma(W_l[v_l(x)]_i + b_l))$. If $O_l = 0$, we have $v_l(x) = h_l(x)$ from the first equation. If $U_l = 0$, we have $h_{l+1}(x) = v_l(x)$ from the second equation. Therefore we have $\tilde{h}_l(x) = \tilde{h}_L(x)$, for any $l \in [L,...,L+L']$.

### B.1.3 Viewing a Small Transformer as a Special Case of a Large Transformer

As demonstrated above, smaller transformers can be embedded into larger transformers with functionality preserved. The smaller transformer architecture can therefore be viewed as a subset of the larger transformer architecture. In this sense, a set of transformers with different sizes and the same pre-training loss found in Section 2 can be viewed as a set of transformers with the same size after the embedding. Note that the training algorithm only finds out the natural larger models, instead of the larger models which are embedded from the smaller models. This indicates that the implicit bias of the optimizer can interact with the model architecture. The implicit bias drives the model toward flat minima on both larger models and smaller models. The smaller transformer architecture is a subset of the larger transformer architecture, thus the flattest minima found with a larger transformer is flatter than the minima found with a smaller transformer. (See Figure 6).

## C   OMITTED PROOFS IN SECTION 3

*Proof of Lemma 3.2.*  We first recall loss

$$L(\theta) = \mathbb{E}_{x,t}[-\log [f_\theta(x_{-t})]_{x_t}] = \mathbb{E}_{t,x_{-t}}\mathbb{E}_{x_t|t,x_{-t}}[-\log [f_\theta(x_{-t})]_{x_t}].$$

Note that conditioned on any $x_{-t}, t$, it holds that

$$\mathbb{E}_{x_t|t,x_{-t}}\left[-\nabla_\theta^2\log [f_\theta(x_{-t})]_{x_t}\right]$$

$$=\mathbb{E}_{x_t|t,x_{-t}}\left[-\frac{\nabla_\theta^2[f_\theta(x_{-t})]_{x_t}}{[f_\theta(x_{-t})]_{x_t}}\right]+\mathbb{E}_{x_t|t,x_{-t}}\left[\frac{\nabla_\theta[f_\theta(x_{-t})]_{x_t}(\nabla_\theta[f_\theta(x_{-t})]_{x_t})^\top}{[f_\theta(x_{-t})]_{x_t}^2}\right]$$

$$=0+\mathbb{E}_{x_t|t,x_{-t}}\left[\nabla_\theta\log[f_\theta(x_{-t})]_{x_t}(\nabla_\theta\log[f_\theta(x_{-t})]_{x_t})^\top\right],$$

where in the last step, we use the assumption that $\theta \in \Gamma$, that is, for all $x, t$, $f_\theta(x_{-t}) = \Pr(\cdot\,|\,x_{-t})$, which implies the following

$$\mathbb{E}_{x_t|t,x_{-t}}\left[-\frac{\nabla_\theta^2[f_\theta(x_{-t})]_{x_t}}{[f_\theta(x_{-t})]_{x_t}}\right]=-\sum_{x_t=1}^{c}\nabla_\theta^2[f_\theta(x_{-t})]_{x_t}=-\nabla_\theta^2\sum_{x_t=1}^{c}[f_\theta(x_{-t})]_{x_t}=-\nabla_\theta^2 1=0.$$

Since $\theta$ is a global minimizer of $L$, we have that $\nabla L(\theta) = \mathbb{E}_{t,x}\nabla_\theta\log[f_\theta(x_{-t})]_{x_t} = 0$. Therefore, we have that

$$\Sigma(\theta) = \mathbb{E}_{t,x}\left[\nabla_\theta\log[f_\theta(x_{-t})]_{x_t}(\nabla_\theta\log[f_\theta(x_{-t})]_{x_t})^\top\right]$$

$$-\mathbb{E}_{t,x}\nabla_\theta\log[f_\theta(x_{-t})]_{x_t}(\mathbb{E}_{t,x}\nabla_\theta\log[f_\theta(x_{-t})]_{x_t})^\top$$

$$=\mathbb{E}_{t,x_{-t}}\mathbb{E}_{x_t|t,x_{-t}}\left[\nabla_\theta\log[f_\theta(x_{-t})]_{x_t}(\nabla_\theta\log[f_\theta(x_{-t})]_{x_t})^\top\right]$$

$$=\mathbb{E}_{t,x_{-t}}\nabla_\theta^2\left(\mathbb{E}_{x_t|t,x_{-t}}[-\log [f_\theta(x_{-t})]_{x_t}]\right)$$

$$=\nabla^2 L(\theta),$$

which completes the proof. $\qquad\qquad\qquad\qquad\qquad\qquad\qquad\qquad\qquad\qquad\square$

## D  PRACTICAL IMPLICATIONS

**Pre-training Algorithms.**  While we focus on the saturation regime in the paper as a controlled way to compare models with the same pre-training loss, the overall takeaway is that the implicit bias of the training algorithms matters for downstream performance (no matter whether we are in the saturation regime or not). Moreover, understanding the implicit biases needed for downstream performance may also lead to better training methods (instead of better evaluation methods) that might encourage the correct biases more strongly. Therefore, a practical direction is to design better pre-training algorithms with more favorable biases which can lead to better downstream performance than AdamW and SGD.

**Better Metrics for Language Models.**  In common practice, the validation pre-training loss is used to monitor the training process (Brown et al., 2020; Zhang et al., 2022a) and compare different models (Hernandez et al., 2021). However, Saunshi et al. (2022); Tay et al. (2021) show that pre-training loss is not necessarily correlated with downstream performance when comparing different architectures. We further show that pre-training loss may not always be a reliable indicator even for the same architecture. While downstream tasks could be used as a proxy metric for evaluation, the main issue is that large language models are trained to be general / multi-purpose models where the space of downstream tasks is large and unknown during the time of pre-training. Thus from a fundamental standpoint, it is beneficial to design a more reliable indicator that is agnostic to downstream tasks.

**Explicit regularization.**  We show that implicit bias, especially the implicit bias of flatness matters for downstream performance in language modeling. Leveraging the implicit bias to design better explicit regularization in language modeling is also an important direction. Bahri et al. (2021) show explicit flatness regularization with SAM (Foret et al., 2020) can boost downstream performance when applying to downstream tasks themselves and the intermediate stages between pre-training and fine-tuning, but they did not study this on pre-training, partly because SAM is not efficient enough for pre-training (SAM requires back prop for 2 times per step, and more steps to reach the same level of pre-training loss (Foret et al., 2020)).

# E RESULTS OF 15% MASK RATE

As we mentioned in Section 2 and Section A.2, the true conditional probability of MLM $\Pr(x_t \,|\, x_{-t})$ can be computed from the joint probability

$$\Pr(x_t = c \,|\, x_{-t}) = \frac{\Pr(x_t = c, x_{-t})}{\sum_{c \in W} \Pr(x_t = c, x_{-t})},$$

and the joint probability can be computed efficiently with knowledge of the generative algorithms. The number of computation of the joint probability is the size of $W$.

However, when we have multiple masks, the time complexity of computing the true conditional probability can be extremely large. For example, if we have two masks in one sentence,

$$\Pr(x_t = c, x_{t'} = c' \,|\, x_{-t,t'}) = \frac{\Pr(x_t = c, x_{t'} = c', x_{-t,t'})}{\sum_{c,c' \in W} \Pr(x_t = c, x_{t'} = c', x_{-t})}.$$

Now to compute the denominator, we will need to compute the joint probability for $|W|^2$ times. In general, the time complexity of computing the true MLM conditional probability is exponential in the number of masks.

Still, we can pre-train the model with 15% mask rate and evaluate the loss with one mask per sentence. The results on PCFG is provided in Figure 11. Although in pre-training we use 15% mask rate, the validation loss evaluated with one mask per sentence does not change much compared with training with one mask per sentence. The downstream performance does not change significantly with 15% mask rate either. The conclusion of Section 2 and Section 4 still holds.

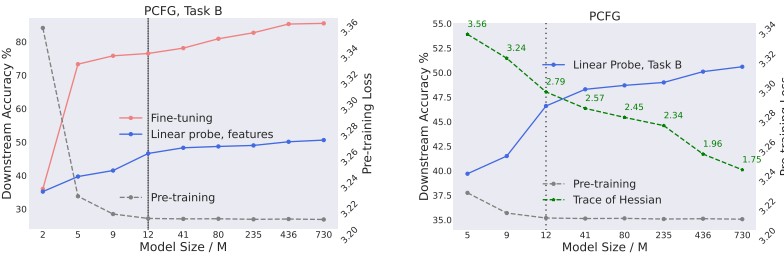

Figure 11: The result of pre-training with 15% mask rate and evaluating with 1 mask per sentence on PCFG. Left: Larger models perform better downstream than smaller models, even with almost the same pre-training loss. Right: The trace of Hessian correlates with downstream performance for models with different sizes and almost the same pre-training loss.

# F  OMITTED PROOFS IN SECTION 5

## F.1  OMITTED PROOFS OF THEOREM 5.1

Recall that the loss function of MLM is $L(\psi, u) = \mathbb{E}_{x \sim P}[(f_{\psi,u}(x) - g^*(x))^2]$. In downstream adaptation, we have access to a finite dataset $\{x^{(i)}\}_{i=1}^n$ sampled i.i.d. from $P_{\mathrm{ds}}$. The training loss is $\widehat{L}^{P_{\mathrm{ds}}}(\psi, u) = \frac{1}{n} \sum_{i=1}^n [(f_{\psi,u}(x^{(i)}) - g^*(x^{(i)}))^2]$, and the population loss for the downstream task is $L^{P_{\mathrm{ds}}}(\psi, u) = \mathbb{E}_{x \sim P_{\mathrm{ds}}}[(f_{\psi,u}(x) - g^*(x))^2]$.

*Proof of Theorem 5.1.* We first calculate the trace of Hessian of the pre-training loss and then derive a lower bound for it in Lemma F.1. We then show that the lower bound can be achieved only if the output of the attention are in one direction for all the downstream input in Lemma F.3. This translates to constant sample complexity for the downstream task.

**Lemma F.1.** *Denote by $h_{Q,K}(x) = \sum_{j=1}^T a_j x_j$ the output of the attention head. In the setting of Theorem 5.1, $I_+ = \{i \in [m] \mid u_i > 0\}$ and $I_- = \{i \in [m] \mid u_i < 0\}$. The trace of Hessian can be lower bounded,*

$$\mathrm{Tr}[\nabla_\psi^2 L(\psi, u)] + \mathrm{Tr}[\nabla_u^2 L(\psi, u)] \geq \sqrt{\frac{4}{T}},$$

*where the lower bound is achieved if and only if the following conditions are satisfied,*

$$\forall\, i \in [m], x \in \{x \mid V_i^\top h_{Q,K}(x) > 0\}, \quad V_i^\top h_{Q,K}(x) = |u_i| \|h_{Q,K}(x)\|_2, \tag{3}$$

$$\forall x, i \in I_+, i' \in I_-, \quad V_i^\top x V_{i'}^\top x \leq 0, \tag{4}$$

$$\forall x, j \in [T], \quad a_j = \frac{1}{T}. \tag{5}$$

Denote by $D_+ = \{x \mid y = 1\}$ and $D_- = \{x \mid y = -1\}$. $I_x$ is the set of index of neurons which is activated on $x$, $I_x = \{i \in [T] \mid V_i^\top x > 0\}$. By the condition in equation 5, the attention is taking the average of $[x_t]_{t=1}^T$. We can map $D_+$ and $D_-$ to the feature space, $H_+ = \{h_{Q,K}(x) \mid y = 1\}$ and $H_- = \{h_{Q,K}(x) \mid y = -1\}$. We first show that one neuron cannot be activated on inputs from both $D_+$ and $D_-$, and all non zero neuron has to be activated on some input. Also note that a neuron cannot be activated on no input, unless the weight is $0$.

**Fact F.2.** *(1) $\forall x \in D_+$, $I_x \subseteq I_+$. Similarly we have $\forall x \in D_-$, $I_x \subseteq I_-$. (2) Suppose $V_i \neq 0$, then there exists $h \in H_+ \cup H_-$, $V_i^\top h > 0$.*

*Proof of Fact F.2.* (1) Otherwise, suppose $j \in I_x \cap I_-$, since $y = \frac{1}{m} \sum_{i=1}^m u_i \sigma(V_i^\top h_{Q,K}(x)) > 0$, there has to be $j' \in I_x \cap I_+$, which contradicts the condition in equation 4. (2) Suppose $v^\top h \leq 0$ for all $h \in H_+ \cup H_-$. Then we have $v^\top h = 0$ for all $h$, since $v^\top h < 0$ indicates $v^\top(-h) > 0$, and $-h$ belongs to the support of $P$ due to the symmetry of the distribution. However, in Lemma F.5, we show that the matrix stacking all input together has full row rank, thus $v$ has to be $0$, leading to a contradiction. $\square$

We have the following lemma characterizing the solutions achieving all the qualities in Lemma F.1. Intuitively, all the neurons can be divided into two sets, and each input can only activate neurons in one of the sets, leading to no cancellation between activated neurons. This holds because of equation 4 and the properties of the input distribution.

**Lemma F.3.** *Suppose $Q, K, V$ satisfy the equality in Lemma F.1. For all $i \in I_-$, on downstream data $x$, if $g^*(x) = 1$, we have $V_i^\top h_{Q,K}(x) = c_i > 0$. $c_i$ is a constant which holds for every $x$ if $g^*(x) = 1$. If $g^*(x) = -1$, we have $V_i^\top h_{Q,K}(x) = 0$.*

*For all $i \in I_+$, on downstream data $x$, if $g^*(x) = -1$, we have $V_i^\top h_{Q,K}(x) = c_i > 0$. $c_i$ is a constant which holds for every $x$ if $g^*(x) = 1$. If $g^*(x) = 1$, we have $V_i^\top h_{Q,K}(x) = 0$.*

Now let us consider the downstream task. It suffices to consider the constant vector $c$. If samples satisfying $g^*(x)=1$ and $g^*(x)=-1$ both show up in the downstream dataset, the minimal norm solution $\tilde{u}$ is $\tilde{u}_{I_-} = \frac{mc_{I_-}}{\|c_{I_-}\|_2^2}$, $\tilde{u}_{I_+} = \frac{-mc_{I_+}}{\|c_{I_+}\|_2^2}$, and $\tilde{u}_{[m]\setminus(I_+\cup I_-)}=0$. Then we can verify that

$$f_{\hat{\psi},\tilde{u}}(x) = \frac{1}{m}\left[\sum_{i\in I_+}\frac{-mc_i}{\|c_{I_+}\|_2^2}\sigma(V_i^\top h_{Q,K}(x)) + \sum_{i\in I_-}\frac{mc_i}{\|c_{I_-}\|_2^2}\sigma(V_i^\top h_{Q,K}(x))\right]$$

$$= \frac{1}{m}\left[\sum_{i\in I_+}\frac{-mc_i}{\|c_{I_+}\|_2^2}c_i\mathbb{I}[g^*(x)=-1] + \sum_{i\in I_-}\frac{mc_i}{\|c_{I_-}\|_2^2}c_i\mathbb{I}[g^*(x)=1]\right]$$

$$= \mathbb{I}[g^*(x)=1] - \mathbb{I}[g^*(x)=-1]$$

$$= g^*(x).$$

Therefore, $L^{P_{\text{ds}}}(\hat{\psi},\tilde{u}) = \mathbb{E}_{x\sim P_{\text{ds}}}[(f_{\hat{\psi},\tilde{u}}(x)-g^*(x))^2]=0$, which completes the proof. $\qquad\square$

We first show that when the pre-training loss equals 0, the trace of Hessian equals the square of the norm of the gradient.

**Lemma F.4.** *For any parameters $\theta$, if the pre-training loss $L(\theta)=\mathbb{E}_x[(f_\theta(x)-y)^2]=0$, the trace of Hessian equals the square of the norm of the gradient,*

$$\text{Tr}[\nabla_\theta^2 L(\theta)] = \mathbb{E}[\|\nabla_\theta f_\theta(x)\|_2^2].$$

*Proof of Lemma F.4.* We can express the Hessian as follows.

$$\nabla_\theta^2 L(\theta) = \mathbb{E}_x[\ell'(f_\theta(x),y)\nabla_\theta^2 f_\theta(x)] + \mathbb{E}_x[\frac{1}{2}\ell''(f_\theta(x),y)\nabla_\theta f_\theta(x)\nabla_\theta f_\theta(x)^\top].$$

Since $L(\theta) = \mathbb{E}_x[(f_\theta(x)-y)^2]=0$, we have with probability 1, $\ell'(f_\theta(x),y)=0$ and $\ell''(f_\theta(x),y)=2$ is a constant. $\qquad\square$

*Proof of Lemma F.1.*

$$\text{Tr}[\nabla_\psi^2 L(\psi,u)] + \text{Tr}[\nabla_u^2 L(\psi,u)] = \sum_{\theta\in[Q,K,V,u]}\mathbb{E}[\|\nabla_\theta f_{\psi,u}(x)\|_2^2] \qquad \text{(By Lemma F.4)}$$

$$\geq \mathbb{E}[\|\nabla_V f_{\psi,u}(x)\|_2^2 + \|\nabla_u f_{\psi,u}(x)\|_2^2] \qquad (6)$$

$$= \frac{1}{m}\mathbb{E}\left[\|\sigma(Vh_{Q,K}(x))\|_2^2 + \|h_{Q,K}(x)(\mathbb{I}[Vh_{Q,K}(x)>0]\odot u)^\top\|_2^2\right]$$

$$= \frac{1}{m}\mathbb{E}\left[\sum_{i=1}^m\sigma(V_i^\top h_{Q,K}(x))^2 + \|h_{Q,K}(x)\|_2^2\mathbb{I}[V_i^\top h_{Q,K}(x)>0]u_i^2\right]$$

$$\geq \frac{2}{m}\mathbb{E}\left[\sum_{i=1}^m\sigma(V_i^\top h_{Q,K}(x))\|h_{Q,K}(x)\|_2|u_i|\right] \qquad (7)$$

$$\geq \frac{2}{m}\mathbb{E}\left[\|h_{Q,K}(x)\|_2\left|\sum_{i=1}^m\sigma(V_i^\top h_{Q,K}(x))u_i\right|\right] \qquad (8)$$

$$= 2\mathbb{E}\left[\|h_{Q,K}(x)\|_2|f_{\theta,u}(x)|\right]$$

$$\geq \sqrt{\frac{4}{T}}. \qquad (9)$$

The equality in step 6 is achieved if and only if the gradient of $Q$ and $K$ is 0. Equation (7) is from AM-GM, and the equality is achieved iff

$$V_i^\top h_{Q,K}(x) = |u_i|\|h_{Q,K}(x)\|_2 \quad \forall i\in[m], x\in\{x\,|\,V_i^\top h_{Q,K}(x)>0\}.$$

The equality in step 8 is achieved iff on all input, there is no cancellation between activated neurons,

$$\forall x, i\in I_+, i'\in I_-, \quad V_i^\top x V_{i'}^\top x \leq 0.$$

Since the attention score $a_j$ satisfies $a_j > 0$ and $\sum_{j=1}^{T} a_j = 1$, and all embeddings $x_t$ in one masked sentence are orthogonal to each other with norm 1, we have $\|h_{Q,K}(x)\|_2 \geq \frac{1}{\sqrt{T}}$. The equality is achieved iff $a_j = \frac{1}{T}$ for all $x$ and all $j \in [T]$. $\qquad \square$

*Proof of Lemma F.3.* Suppose $V_i$ is a neuron with $i \in I_-$. Then there exists $h \in H_-$, $V_i^\top h > 0$. Without loss of generality, suppose the masked position in $h$ is 1, i.e. $h_1 = 0$, $h_2 = 1$. Now let us consider the components in $V_i$ corresponding to the input positions and the mask positions separately. $V_i^{(c)} = [V_{i,1}, V_{i,2}, ..., V_{i,T}]$ and $V_i^{(p)} = [V_{i,T+1}, V_{i,T+2}, ..., V_{i,2T}]$.

We claim that $V_i^{(c)}{}_{2:T} = c\mathbf{1}$ for some $c > 0$ and $V_i^{(p)}{}_1$ is either 0 or $c$. To prove this, consider $\tilde{h}$, which is only different from $h$ on the mask, $h - \tilde{h} = 2e_2$. Also consider $-h$ and $-\tilde{h}$. Due to the symmetry of the distribution, $-h$ and $-\tilde{h}$ are in $H_+$. By Fact F.2, $V_i^\top(-h) \leq 0$ and $V_i^\top(-\tilde{h}) \leq 0$. $V_i^\top(-h)$ cannot be 0, because this will leads to $V_i^\top h = 0$.

**Case 1.** $V_i^\top(-h) < 0$ and $V_i^\top(-\tilde{h}) < 0$. We have $V_i^\top h = V_i^\top \tilde{h} > 0$, indicating that $V_i^\top(h - \tilde{h}) = 2V_i^{(p)}{}_1 = 0$. Now we show that $V_i^{(c)}{}_{2:T} = c\mathbf{1}$. Due to the condition of equation 3, we know that for any $h \in H_-$ masked on the first token, either $V_i^\top h = 0$ or they equal to the same positive value $c$ for all $h$. We claim that $V_i^\top h = c$ for any $h \in H_-$. Otherwise there exist $H_{-,1}$ and $H_{-,0}$, $H_{-,0} \cap H_{-,1} = \emptyset$ and $H_{-,0} \cup H_{-,1} = H_- \cap \{h \mid h_2 = \pm 1\}$. $V_i^\top h = c$ for any $h \in H_{-,1}$ and $V_i^\top h = 0$ for any $h \in H_{-,0}$. This cannot happen for $T \geq 6$. By Lemma F.5 we know that the matrix stacking all such $h_{2:T}$ together has full row rank, thus $V_i^{(c)}{}_{2:T} = c\mathbf{1}$ for some $c > 0$ and $V_i^{(p)}{}_1 = 0$.

**Case 2.** $V_i^\top(-h) < 0$ and $V_i^\top(-\tilde{h}) = 0$, which indicates $V_i^\top h = V_i^{(p)}{}_1$. Consider another $h' \in H_-$ which is not equal to $h$ and $h'_2 = 1$. Similarly we can find $\tilde{h}'$, $h' - \tilde{h}' = 2e_2$. Still we have $V_i^\top(-h') < 0$ and $V_i^\top(-\tilde{h}') = 0$, this tells us $V_i^\top h' = V_i^\top h$, due to the condition of equation 3. Applying this to different $h'$s, we have that $V_i^\top h$ equals the same positive value for all $h \in H_-$ and the masked position is 0. By Lemma F.5 we know that the matrix stacking all such $h_{2:T}$ together has full row rank, thus $V_i^{(c)}{}_{2:T} = c\mathbf{1}$ for some $c > 0$ and $V_i^{(p)}{}_1 = c$.

We have proved that $V_i^{(c)}{}_{2:T} = c\mathbf{1}$ for some $c > 0$ and $V_i^{(p)}{}_1$ is either 0 or $c$. We continue to show that $V_i^{(c)} = c\mathbf{1}$ for the same $c > 0$ and either $V_i^{(p)}$ is 0 or its coordinates is $\pm c$.

For case 1, consider $h'' \in H_-$ whose masked position is 2, $h''_4 = 1$. Also suppose that $h''_1 = 1$ By Fact F.2, we know that $V_i^\top(-h'') \leq 0$ and $V_i^\top(-\tilde{h}'') \leq 0$, this implies that $-V_i^{(c)}{}_1 \leq V_i^{(p)}{}_2 \leq V_i^{(c)}{}_1$. Applying the same argument above, we know that either $V_i^{(p)}{}_2 = 0$ or $V_i^{(p)}{}_2 = \pm V_i^{(c)}{}_1$, otherwise both $V_i^\top h'' > 0$ and $V_i^\top \tilde{h}'' > 0$ hold, and $V_i^\top h'' \neq V_i^\top \tilde{h}''$, contradicting condition in equation 3. If $V_i^{(p)}{}_2 = V_i^{(c)}{}_1$, from equation 3 we know $V_i^\top h'' = V_i^\top h$, which indicates $V_i^{(p)}{}_2 = V_i^{(c)}{}_1 = \frac{c}{2}$. In this case we can find another $h'''$ whose masked position is 2, $h'''_4 = 1$ but $h'''_1 = -1$. Then $V_i^\top h''' \neq V_i^\top h$, contradicting equation 3. Thus $V_i^{(p)}{}_2 \neq V_i^{(c)}{}_1$. Similarly $V_i^{(p)}{}_2 \neq -V_i^{(c)}{}_1$. The only possible situation is $V_i^{(p)}{}_2 = 0$. Applying the argument in this paragraph to other masked position, we have $V_i^{(p)} = 0$. Since $H_-$ is invariant under permutation, $V_i^{(c)}{}_1 = c$, and $V_i^{(c)} = c\mathbf{1}$.

For case 2, exactly the same argument as the above paragraph with the same $h''$ and $h'''$ shows that the coordinates of $V_i^{(p)}$ is $\pm c$.

Therefore, we have shown that for any $i \in I_-$, $V_i^{(c)} = c\mathbf{1}$ for same $c > 0$. The symmetry of distribution immediately tells us for any $i \in I_+$, $V_i^{(c)} = c\mathbf{1}$ for $c < 0$.

On the downstream distribution $P_*$, since there is no masked token, only $V_i^{(c)}$ is working. Since $V_i^{(c)} = c\mathbf{1}$ always holds, we complete the proof. $\qquad \square$

**Lemma F.5.** *Suppose $M \in \mathbb{R}^{(2k+1) \times (2k+1)}$ is a matrix composed of $\pm 1$. The first row of $M$ is $[\underbrace{1,...,1}_{k+1 \text{ 1's}}, \underbrace{-1,...,-1}_{k \text{ -1's}}]$. Define a permutation $\rho(1) = 2$, $\rho(2) = 3 ... \rho(2k+1) = 1$. For all $i \geq 2$, $M_{i,\rho(j)} = M_{i-1,j}$. Then the rank of $M$ is $2k+1$.*

*Proof of Lemma F.5.* Note that $M_i + M_{\rho^{k+1}(i)} = 2e_i$ for all $i \in [2k+1]$, which means we can express the orthonormal basis as linear combination of the rows in $M$. Therefore, the rank of $M$ is $2k+1$. $\square$

## F.2 THE EXISTENCE OF RANDOM FEATURE SOLUTIONS F.6

**Theorem F.6.** *Suppose $m \geq \tilde{O}(2^T T^3 \epsilon^{-2})$, and $V_i \sim \mathcal{N}(0, T I_{2T})$ for all $i \in [m]$. With probability at least $1 - \delta$ over $V$, there exists $\psi', u'$, satisfying $L(\psi', u') \leq \epsilon$ and $\|u'\|_2^2 \leq O(T^2 \delta^{-1})$.*

*Proof of Theorem F.6.* Since the number of possible input in pre-training is finite, we can invoke Lemma 9 in Bai & Lee (2020) to show that random Gaussian features can fit the pre-training task.

**Lemma F.7** (Lemma 9 in Bai & Lee (2020)). *Suppose $\|h\|_2 = \sqrt{\frac{1}{T}}$, $v \sim \mathcal{N}(0, T I_{2T})$. There exists a random variable $a(v)$ such that*

$$\mathbb{E}[\sigma(v^\top h)a] = -\sqrt{T}\mathbf{1}^\top h$$

*and $a$ satisfies $\mathbb{E}_v[a^2] = O(T^2)$.*

Consider $Q = K = 0$. In this case we have $h_{Q,K}(x) = \frac{1}{T}\sum_{j=1}^T x_j$. Also note that for the pre-training task, $y = -\sqrt{T}\mathbf{1}^\top h_{Q,K}(x)$ for all $x$. Since $x_j$ are norm 1 orthogonal to each other, we have $\|h_{Q,K}(x)\|_2 = \frac{1}{T}$ for all $x$. Now we can show that $V_i \sim \mathcal{N}(0, T I_{2T})$, $i \in [m]$ independently is the random feature solution which can solve the pre-training task.

Suppose $g(h) = \frac{1}{m}\sum_{r=1}^m \sigma(V_r^\top h)a(V_r)$, and $g_{(R)}(h) := \frac{1}{m}\sum_{r=1}^m \sigma(V_r^\top h)a(V_r)\mathbb{I}(\|V_r\|_2 \leq \sqrt{T}R)$. $R$ is large enough such that $\Pr\left(\sup_{r\in[m]}\|V_r\|_2 \geq \sqrt{T}R\right) \geq 1 - \delta/2$. We have $g(h) = g_{(R)}(h)$ on this event. Let $g_{(R)}^*(h)$ be the truncated version of $g^*(h) = -\sqrt{T}\mathbf{1}^\top h$, $g^*(h)_{(R)} = \mathbb{E}_v[\sigma(v^\top h)a(v)\mathbb{I}(\|v\|_2 \leq \sqrt{T}R)]$. We have

$$\mathbb{E}_V[(g(h) - g^*(h)_{(R)})] \leq \frac{1}{m}\mathbb{E}_v[\sigma(v^\top h)^2 a(v)^2 \mathbb{I}(\|v\|_2 \geq \sqrt{T}R)] \leq C\frac{R^2 T^2}{m}.$$

By Chebyshev and a union bound, we have

$$\Pr\left(\max_h |g(h) - g^*(h)_{(R)}| \geq t\right) \leq C\frac{n(h)R^2 T^2}{mt^2}.$$

For $t = \frac{\epsilon}{2}$, we have $m \geq n(h)R^2 T^2 \epsilon^{-2}$.

$$|g_{(R)}^*(h) - g^*(h)| = \mathbb{E}_v[\sigma(v^\top h)a(v)\mathbb{I}(\|v\|_2 \geq \sqrt{T}R)]$$

$$\leq \mathbb{E}_v[a(v)^2]^{\frac{1}{2}}\mathbb{E}_v[\sigma(v^\top h)^4]^{\frac{1}{4}}\Pr\left(\|v\|_2 > \sqrt{T}R\right)^{\frac{1}{4}}$$

$$\leq CT\Pr\left(\|v\| > \sqrt{T}R\right)^{\frac{1}{4}}.$$

Choosing $R = \tilde{O}(\sqrt{(T)})$ will make $\Pr\left(\|v\| > \sqrt{T}R\right)^{\frac{1}{4}} \leq \frac{c\epsilon}{T}$. Also note that $n(h) = (T/2 - 1)\binom{T}{T/2+1}$. Thus $m \geq \tilde{O}(2^T T^3 \epsilon^{-2})$ suffices. $\qquad\square$

