# OpenReview forum: "Same Pre-training Loss, Better Downstream: Implicit Bias Matters for Language Models"
_ICLR.cc/2023/Conference — Submitted to ICLR 2023_

### Official Review · Reviewer_mwuY · 2022-10-22

**Confidence:** 3
**Correctness:** 3
**Technical Novelty And Significance:** 2
**Empirical Novelty And Significance:** 2
**Recommendation:** 6

**Clarity, Quality, Novelty And Reproducibility:**

Although this paper tackles interesting research question, it is unclear that flatter minima actually leads to better generalizations since it does not compare relevant baselines and lack of justification. Please see detain in strength and weakness section.

**Strength And Weaknesses:**


## Strength
- The paper explores the interesting and practical research question --- correlation between pre-training loss and generalization performance of downstream tasks.


- In the controlled experiments, the authors empirically show the correlation between flatness and generalization. Even though language model achieves optimal pre-training loss, we can improve generalization performance of downstream task as we continue pre-training, which results in flatter minima.

- Authors provide some theoretical analysis that connects flatter solution to the pre-training loss and generalization performance of downstream tasks in the limited settings.
---
Please address my following concerns. If they are properly addressed, I am willing to raise my score.
## Weakness

- My expertise not learning theory, but in my humble opinion, what the authors claims is too bold. They claim that they "prove that SGD with standard mini-batch noise implicitly prefers flatter minim in language models", but it is limited to the special case ---  Dyck language experiment which is far from practical scenario.  It would be better to tone down the claim.

- I am a bit confused about the correlation between flatness and generalization. For example, Dinh et al., [1] show that we can build equivalent models but with sharper minima, which might contradict to  what the authors claim, specifically theorem 5.1. However, the authors did not properly tackle this issue even in the related work section.


- It is not clear how the size of models and flat minima is related to each other. The authors claim that smaller transformer architecture is a subset of the larger transformer architecture family, but it does not make sense. If we increase the width of transformer or increase the vocabulary size, the smaller model cannot be a special case of larger model. It would be better to explain why larger model converges to flatter minima than smaller one.

- In the experiments, it is unclear why the authors compare vanilla sgd training with adversarial training. I think adversarial weight perturbation [2] is more relevant baseline since it explicitly regularize the flatness of weight loss landscape.

- Minor typo: In page 4, (2) training for different number of steps -> (1) training for different number of steps


## Questions

- Regarding the flat minima, what happens if we explicitly enforce the language model to converge flatter minima? For example, we can explicitly regularize the trace of Hessian to be small or use SAM [3] optimizer to enforce such regularization.

- Why is the cross-entropy loss is typically non-zero at the global minimizers?

- Why do you assume that we only mask a single token in a sentence for masked language model? In practice, we usually mask 15% of tokens  of a sentence (e.g. BERT [4], RoBERTA [5]).

## References
[1] Dinh, Laurent, et al. "Sharp minima can generalize for deep nets." International Conference on Machine Learning. PMLR, 2017.

[2] Wu, Dongxian, Shu-Tao Xia, and Yisen Wang. "Adversarial weight perturbation helps robust generalization." Advances in Neural Information Processing Systems 33 (2020): 2958-2969.

[3] Foret, Pierre, et al. "Sharpness-aware Minimization for Efficiently Improving Generalization." International Conference on Learning Representations. 2021.

[4] Devlin, Jacob, et al. "BERT: Pre-training of Deep Bidirectional Transformers for Language Understanding." Proceedings of the 2019 Conference of the North American Chapter of the Association for Computational Linguistics: Human Language Technologies, Volume 1 (Long and Short Papers). 2019.


[5] Liu, Yinhan, et al. "Roberta: A robustly optimized bert pretraining approach." arXiv preprint arXiv:1907.11692 (2019).



**Summary Of The Paper:**

This paper questions the common sense that correlation between pre-training validation loss such as masked language modeling loss and generalization performance of downstream tasks after fine-tuning. It empirically shows that the models with the same pre-training loss can achieve varying test accuracy on downstream tasks. Instead, the models converged to flat minima obtain better test accuracy.

**Summary Of The Review:**

As previously mentioned, the authors do not properly tackle the open question --- flat minima and generalization which might still seem to be controversial topic. Moreover, they need to tone down their claim since their theoretical analysis is limited to very specific controlled experiment which is far from realistic scenario. Thus, I am inclined to reject.

---

> ### Author Response · Authors · 2022-11-11
> **Response to Reviewer mwuY (Part 3)**
>
> >**Q7:** Why is the cross-entropy loss typically non-zero at the global minimizers?
>
> That’s because in language modeling, the label given a masked sentence is not deterministic. For example, "_I ____ coffee._" can be filled in with drink, have, like… The label follows a distribution, corresponding to the true conditional probability we mentioned in Section 2 of the paper. No matter what the prediction is, the cross entropy loss cannot be zero.
>
> Another perspective is that the cross entropy is the sum of the KL divergence (predicted || true) and the entropy of the true conditional probability. Since KL divergence is non-negative, and the entropy of the true conditional probability is **strictly positive** because the label is not deterministic, the cross entropy loss cannot be 0.
>
> In fact, the minimal possible cross-entropy loss for language modeling is the entropy of the language.
>
> >**Q8:** Why do you assume that we only mask a single token in a sentence for masked language model?
>
> As we mentioned in the experimental setup in Section 2 and Section A, with the knowledge of the generative models (PCFG, HMM, and OPT), we can compute the true MLM conditional probability with one mask per sentence. This is crucial to compute the pre-training loss accurately and make sure the pre-training loss of different models are nearly identical. If we allow multiple masks, we can no longer compute the true MLM conditional probability efficiently from the generative models because the time complexity of computing the true MLM conditional probability is _exponential_ in the number of masks for PCFG and OPT. Still, we provide the result of training with 15% mask rate and evaluating the pre-training loss with one mask per sentence in the revision $\underline{\text{Section E}}$. The result is almost the same as pre-training with one mask per sentence. Pre-training with 15% mask rate does not change pre-training loss (evaluated with 1 mask per sentence) and the downstream performance much. The conclusion of Section 2 and Section 4 still holds.
>
> [1] Label noise sgd provably prefers flat global minimizers. NeurIPS 2021
> [2] Sharpness-Aware Minimization Improves Language Model Generalization.
> [3] Sharpness-aware Minimization for Efficiently Improving Generalization.

---

> ### Author Response · Authors · 2022-11-11
> **Response to Reviewer mwuY (Part 2)**
>
> >**Q4:** “The authors claim that smaller transformer architecture is a subset of the larger transformer architecture family, but it does not make sense.”
>
> As we justify in Section B of the submission, the smaller transformer architecture is a subset of the larger transformer architecture family, roughly by replicating the weights of the smaller transformer to add the width and leveraging residual connections to add the depth. This transformation preserves the black-box functionality of the small transformer. For example, to go from a 2 layer model to a 4 layer model, we can introduce two layers of zeros at the end, which ensures that the output of the 4th layer will be the same as the output of the 2nd layer via residual connections. To increase the width of the network from 256 to 512, we can replicate the weight of the small transformers and concatenate them as the weight of the large transformer.
>
> In more detail: Suppose the small transformer architecture is $\mathcal{F}$, and the large transformer architecture is $\mathcal{G}$. The smaller transformer architecture is a subset of the larger transformer architecture family, because for each small transformer $f\in\mathcal{F}$, we can construct a larger transformer $g\in\mathcal{G}$ (with more layers and increased width) whose functionality is the same as the small transformer, i.e. $f=g$. We provide the method in the revision Section B.1. We also include the code in [this link](https://anonymous.4open.science/r/iclr2023_2440-DFBD/from_small_to_large_add_layers.ipynb). In this way, $\mathcal{F}\subset \mathcal{G}$. Let’s then consider the optimization problems:
>
> $\text{minimize}_{g \in \mathcal{G}} \text{Tr}(\nabla^2 L(g))$  s.t. $L(g) \le \epsilon$.
>
> $\text{minimize}_{f \in \mathcal{F}} \text{Tr} ( \nabla ^2 L(f))$   s.t. $L(f) \le \epsilon$.
>
> Since $\mathcal{F} \subset \mathcal{G}$, we know the result of the second optimization problem is no larger than the first one, as we demonstrate in Figure 4 (right).
>
> >**Q5:** It is unclear why the authors compare vanilla sgd training with adversarial training.
>
>  We compare with the adversarial method to show that there indeed exist models with the same pre-training loss but different downstream performance.
> However, the method we compare with is **not** adversarial training. Instead, it is an adversarial algorithm using bi-level optimization which **makes the model not transferable to the downstream datasets**, and keeps the performance on the pre-training task. We provided the details of this algorithm in Section A.4.
>
> >**Q6:** What happens if we explicitly enforce the language model to converge flatter minima?
>
> We agree that experiments using flatness regularization on real datasets could be an interesting direction – however, our main goal is to show that implicit bias of pre-training algorithms has a role in downstream performance, and we leave explicit regularization in language modeling to future work. In addition, we also note encouraging signs in the literature that flatness regularization may also work in language modeling (instead of only for supervised learning as in SAM [3]). [2] shows explicit flatness regularization with SAM can boost downstream performance. [2] applies SAM to downstream tasks themselves and the intermediate stages between pre-training and fine-tuning, but they did not study this on pre-training, partly because SAM is not efficient enough for pre-training (requires back prop for 2 times per step, and more steps to reach the same level of pre-training loss).

---

> ### Author Response · Authors · 2022-11-11
> **Response to Reviewer mwuY (Part 1)**
>
> Thanks for taking the time reading the paper and providing detailed comments. We first give a general overview of the paper and address the questions below.
>
> General overview:
> - As noted by Reviewer 6MPN, we are the first to find out with experiments the existence of implicit bias in language modeling (Section 2). Models with the same pre-training loss can have different downstream performance, and the reason for the difference is the pre-training algorithm.
> - After discovering the implicit bias in LM, we study the role of implicit bias in LM. We show that when the pre-training loss reaches minimum value, SGD with standard mini-batch noise implicitly prefers flatter minima in language models. This result **does not** require assumptions on specific input distributions, but **a property of SGD** itself.
> - In Section 4, we further show that the flatness of the models we found in Section 2 correlates well with downstream performance while the pre-training loss cannot correlate with downstream performance.
> - In Section 5, we show flatness regularization induced by the implicit bias can identify transferable solutions in a synthetic language setting.
>
> > **Q1**: “I am a bit confused about the correlation between flatness and generalization” and the relation to Dinh et al.
>
> We clarify that we **do not** study the relationship between flat minima and generalization **on the same distribution**, which is the supervised learning setting of Dinh et al. and follow-up works. Instead, we study the role of **implicit bias in language modeling** on performance on downstream tasks from **different distributions**. We will clarify this further in the related works. We find a strong empirical correlation between flatness and downstream performance, and Theorem 5.1 shows a theoretical connection on a restricted scenario. Some previous works have noted that flatness regularization during fine-tuning can help generalization in language modeling [2], but they do not study flatness during pre-training. We thank the reviewer for bringing up this confusion.
>
> >**Q2**: “ For example, Dinh et al. show that we can build equivalent models but with sharper minima, which might contradict to what the authors claim, specifically theorem 5.1. “
>
> Our results do not contradict Dinh et al. The conclusion of Dinh et al. is that there exist handcrafted sharp models that generalize well, that is, generalization doesn’t imply flatness. Our message is that flat models (with the minimal pre-training loss) typically generalize well, that flatness implies transferability to downstream tasks; and we prove this in Theorem 5.1 for the synthetic language setting.
>
>
> >**Q3**: The authors claim that they "prove that SGD with standard mini-batch noise implicitly prefers flatter minima in language models", but it is limited to the special case --- Dyck language experiment which is far from practical scenario.
>
> We believe there is a misunderstanding here: we clarify that the results in Section 3 **do not** rely on the Dyck language, and it applies to almost any family of parameterized models. In ($\underline{\text{Theorem 3.3}}$), we prove that SGD with standard mini-batch noise implicitly prefers flatter minima in language models, when the pre-training loss reaches its minimum value. This result does not require assumptions on the Dyck language. This is similar to [1] ”Label noise SGD probably finds flatter minima”, which also does not rely on specific input distribution. We also verify this theory with experiments in Figure 3 on PCFG, HMM, and OPT-generated datasets.
>
> In Section 5, we show that in a synthetic language similar to Dyck, flatter pretrained models are more generalizable to downstream tasks. This claim relies on the specific synthetic language, and we state this clearly in the introduction and Section 5.
>
> The overall takeaway is that the implicit bias of the training algorithms matters for downstream performance. Therefore, a practical direction is to design better pre-training algorithms with more favorable biases which can lead to better downstream performance than AdamW and SGD.

---

> ### Author Response · Authors · 2022-11-18
> **Further Discussion**
>
> Dear reviewer, may we ask if you could respond to our comments? In our response below, we clarified the difference between our work and Dinh et al. and why their work *does not* contradict our claim. We justified how we can construct a large transformer with the same functionality as a small transformer. We provide the details in the revision Section B.1 and attached the link to the code. We also explained why we use one mask per sentence in the paper and provided the results of training with 15% mask rate in Section E. Please let us know if you have other questions or concerns. Thank you!

---

> ### Comment · Reviewer_mwuY · 2022-11-19
> **Thank you for clarification.**
>
> Sorry for the late reply and thank you for the clarification. I skimmed through it and it seems like most of my concerns are addressed. Highly likely I will raisr my score, but I did not look into the detail yet. It will take some time to go through the revision and the comment.

---

> ### Comment · Reviewer_mwuY · 2022-11-25
> **Update to my initial review**
>
> First of all, thank you for correcting my misunderstanding. So I raise the score to 6. I have some following questions.
>
> - Regarding my [Q3],  sorry for miscommunication. What I wanted to ask is theoretical analysis about the relationship between flatter local minima of language model loss and generalization performance of downstream tasks. Theorem 3.3 shows that SGD with mini-batch noise prefers flatter local minima, but it seems to say nothing about generalization performance of downstream tasks. Theorem 5.1 indeed shows how flatter minima leads to better generalization, but it may not be applicable to general setting other than Dyck language experiment.
>
> - As I have already asked at my initial review, can we say that a  transformer model with smaller vocabulary size belongs to the same family of transformers with larger vocabulary? If so, how can we handle different tokenizations and vocabulary?

---

> > ### Author Response · Authors · 2022-11-25
> > **Thanks for the reply!**
> >
> > We believe a full theoretical explanation of the relationship between flatter minima of language model loss and downstream performance is very challenging—it still remains an open question why flatter models have better generalization in supervised learning despite that researchers generally believed that the key implicit bias of SGD is the flatness in supervised learning [4,5]. Thus it’s even more challenging in the pre-training and downstream adaptation setting with the current theoretical tools. To make an initial step toward this direction, we provide Theorem 5.1 in a synthetic language. We hope that our work encourages the field to look further into the relationship between flatness (and other possible implicit biases) and downstream performance.
> >
> > We always consider a fixed tokenization for a pre-training dataset, because we want to keep the pre-training loss of different models directly comparable. If we allow different tokenization and vocabulary size, the pre-training MLM objective of the models will become different and we cannot compare their pre-training loss fairly without using techniques like bits-per-byte (which results in further complications).
> >
> > [4] Label noise sgd provably prefers flat global minimizers. Neurips 2021
> >  [5] What Happens after SGD Reaches Zero Loss? --A Mathematical Framework ICLR 2022

---

### Official Review · Reviewer_KN6A · 2022-10-23

**Confidence:** 4
**Clarity, Quality, Novelty And Reproducibility:** clear presentation, but not very orig…
**Correctness:** 3
**Technical Novelty And Significance:** 2
**Empirical Novelty And Significance:** 2
**Recommendation:** 3

**Strength And Weaknesses:**

----Strength----

1. the submission found a correlated trend between the decay of the trace of the Hessian matrix evaluated on the training data and the increase of performance on the downstream tasks.

2. given the above observation, it becomes straightforward to infer that a pre-trained model with parameters at a flat minimum would lead to decent performance on the downstream tasks.

----Weaknesses----


1. the trace of a Hessian matrix in the formulation from this paper is very easy to obtain and one doesn't even need to materialise the matrix. The issue is that the Hessian matrix is evaluated around the global minimum of a model being trained on the pre-training data, and, without any knowledge of the pre-training data, it becomes very difficult to compute the Hessian matrix. The other issue is that it does require many samples to obtain a reasonable estimate of the trace.

2. the trace of the Hessian matrix is only an overview of the magnitude of all eigenvalues. A smaller trace could mean that the minimum might be flatter than other minima with the same loss, but it could also mean that the minimum might be steep in only a few directions, and completely non-informative in others. The latter case wouldn't necessarily give us a strong transferable model, and yet the trace of the Hessian matrix wouldn't be able to tell.

3. maybe I am missing the point and I am happy to be corrected, but the arguments made in this submission or the observations obtained are already well-known empirically, especially in the case of zero-shot and few-shot learning with very large pre-trained transformer models. In addition, the theorems mentioned in the submission don't seem necessary, or, in any case, support the arguments or the observations.

**Summary Of The Paper:**

the submission conducted experiments to study the impact of configurations on the transferability of a pre-trained model through the lens of Hessian matrices, and the studied configurations include the number of training iterations, the training objectives, and the size of a model. Unsurprisingly, the conclusion is that more training iterations, standard training objectives, and larger models would lead to better performance on the downstream tasks.

**Summary Of The Review:**

I don't recommend accepting the submission since the empirical results are well-known and the theorems don't provide theoretical justifications to the observations.

---

> ### Author Response · Authors · 2022-11-11
> **Response to Reviewer KN6A**
>
> We thank the reviewer for the comments! We first provide a general overview of the paper before responding to specific points.
>
> One main contribution of our paper is discovering the existence of implicit bias in language modeling (Section 2). We find out language models with **almost the same pre-training loss**, but different downstream performance. This is achieved by changing factors related to the pre-training algorithms (choice of algorithms, training time, etc). These factors are called implicit bias of the algorithms, which is a well-studied area in supervised learning [1,2,3,4,5,6] but was not explored for self-supervised learning. We then study the role of implicit bias in language modeling in Section 3. We show that when the pre-training loss is close to its minimal value, minibatch SGD implicit biases toward flatter minima in language modeling. We empirically verify the correlation between the flatness and the downstream performance. Note that we also observe that pre-training loss does **not** correlate with downstream performance when the pre-training loss is close to the minimal value.
>
>
>
> > **Q1** “ the arguments made in this submission or the observations obtained are already well-known empirically”
>
> We believe that Reviewer KN6A has a misunderstanding here: while it is well-known that larger models trained on more data have better downstream performance (and better pre-training loss), we conduct controlled experiments in a novel setting where all the models have **near identical pre-training loss**,  but still have different downstream performance. This is the situation where implicit bias of the training algorithm matters in choosing between the set of solutions with the same (optimal) pre-training loss. As noted by Reviewer 6MPN and 8JLN, our results are **indeed novel**, and "there is no existing paper discussing the same topic".
>
> To further illustrate in the example of varying training time: Although it was well-believed that the pre-training loss decides the downstream performance, we find out models with the same pre-training loss can still have *different* downstream performance in Section 2. The difference between these models is the implicit bias of the algorithms. Especially, natural pre-training algorithms with more steps bias towards better downstream performance, with the pre-training loss **unchanged**.
>
> > **Q2:** “In addition, the theorems mentioned in the submission don't seem necessary, or, in any case, support the arguments or the observations.”
>
> Our empirical observation is that implicit bias of the training algorithm becomes the important tiebreaker when the pre-trianing loss is the same minimal possible value for all the models ($\underline{\text{Section 2}}$), and particularly the implicit bias towards flat solutions correlates with downstream performance ($\underline{\text{Section 4}}$). To justify this argument theoretically, we first need to prove show that SGD has an implicit bias towards flat solutions in language modeling ($\underline{\text{Theorem 3.3}}$), and that flatter solutions lead to better downstream performance ($\underline{\text{Theorem 5.1}}$). These correspond well with our empirical findings and fill in important blanks in the implicit bias literature.
>
>
>
> > **Q3:** It becomes very difficult to compute the Hessian matrix. The other issue is that it does require many samples to obtain a reasonable estimate of the trace.
>
> The reviewer is correct with the difficulty to compute the Hessian but we never computed the Hessian directly. (The flatness is the trace of the Hessian, which can be computed much faster than the Hessian itself.)
>
> However, we also respectfully disagree with the reviewer on the sample efficiency of the trace of the Hessian. We can even estimate the trace of Hessian with the sampling technique inspired by $\underline{\text{Lemma 3.2}}$ (detailed in Section B) and [7]. For more details, please see the code we provided in trH.py. This approximation is very efficient: 50 samples for each masked sentence and 10000 masked sentences suffice, which takes less than 5 min on a single Titan-RTX GPU.
>
>
>
> [1] [On the Implicit Bias in Deep-Learning Algorithms](https://arxiv.org/abs/2208.12591) (A survey)
> [2] Implicit bias of gradient descent on linear convolutional networks Neurips 2018
> [3] Gradient descent maximizes the margin of homogeneous neural networks. ICLR 2019
> [4] Label noise sgd provably prefers flat global minimizers. Neurips 2021
> [5] What Happens after SGD Reaches Zero Loss? --A Mathematical Framework ICLR 2022
> [6] Kernel and rich regimes in overparametrized models. COLT 2020
> [7] The Implicit and Explicit Regularization Effects of Dropout

---

> ### Author Response · Authors · 2022-11-21
> **Further Discussion**
>
> Dear reviewer, may we ask if you could respond to our comments? In our response below, we clarified that the language models we find out have **almost the same pre-training loss**, but different downstream performance. This is different from previous works showing that larger models have better downstream performance because their pre-training loss is better. We also clarify on our method to efficiently compute the of trace of Hessian without evaluating the Hessian matrix itself. Please let us know if you have other questions or concerns. Thank you!

---

### Official Review · Reviewer_8JLN · 2022-10-25

**Confidence:** 3
**Clarity, Quality, Novelty And Reproducibility:** The paper is well-written and present…
**Correctness:** 4
**Technical Novelty And Significance:** 3
**Empirical Novelty And Significance:** 3
**Recommendation:** 6

**Strength And Weaknesses:**

This work seeks to better understand an important question in representation learning: how does pretraining performance correlate with downstream performance?

This question is adequately answered in controlled settings where we observe the non-correlation between pretraining loss and downstream performance and the correlation between solution flatness and downstream performance. The investigation is both well-motivated and nicely executed. In addition, a theoretical result is provided in support of the empirical observations.

It is, however, unclear what the practical implications of this work are. First of all, current large language models are not in the saturation regime, and it is hard to estimate when they will be as datasets grow with model size in tandem. Second, while this paper points that pretraining loss is not a reliable indicator of downstream performance, a simple remedy is to evaluate on downstream tasks during pretraining and compare models accordingly, which is likely already done in practice. Finally, this paper does not demonstrate if the insight gleaned in this work can lead to additional “flatness regularization” that induces better downstream performance on real datasets. It is understandable that large-scale experiments are expensive and are not expected, but given the rather empirical motivation of the paper, some validation on real data seems desirable.


**Summary Of The Paper:**

Practitioners tend to gauge the downstream performance of large language models by comparing their pretraining losses. However, this paper shows that when the pretraining loss is near convergence, downstream performance can vary depending on the training methods, despite near identical pretraining loss. While there isn’t necessarily a correlation between the pretraining loss and downstream performance, the “flatness” of the solution, as characterized by the trace of the Hessian, does correlate with downstream performance. This observation is then formalized and proved on a synthetic language task.

**Summary Of The Review:**

Interesting empirical observations and theoretical result. However, more empirical results on real data, especially on how the discovered insight can enable better downstream performance, would strength this work given the rather empirical motivation.

---

> ### Author Response · Authors · 2022-11-11
> **Response to Reviewer 8JLN**
>
> Thanks for the constructive comments! 8JLN states that the paper has “interesting empirical observations and theoretical results” and is “well-motivated and nicely executed”. The main goal of our paper is identifying and understanding the effect of implicit bias in language model pre-training. We believe using the insights of this paper to design better pre-training algorithms is an important direction for future work. We discuss the practical implications below, and incorporate these discussions in the revision (Section D).
>
> >**Q1:**  It is “unclear what the practical implications of this work are” - “current large language models are not in the saturation regime, and it is hard to estimate when they will be as datasets grow with model size in tandem.”
>
> While we focus on the saturation regime in the paper as a controlled way to compare models with the same pretraining loss, the overall takeaway is that the implicit bias of the training algorithms matters for downstream performance (no matter whether we are in the saturation regime or not). Therefore, a practical direction is to design better pre-training algorithms with more favorable biases which can lead to better downstream performance than AdamW and SGD.
>
> Moreover, the behavior in the saturation regime is very important and interesting. The perplexity of GPT-3 is already amazingly small (< 1.8). It would be better to be more prepared so that if the newer generations of LLMs do reach the saturation regime, then the researchers who develop those models can get indicators from our work.
>
>
> >**Q2**  “While this paper points out that pretraining loss is not a reliable indicator of downstream performance, a simple remedy is to evaluate on downstream tasks during pre-training and compare models accordingly, which is likely already done in practice.”
>
> While downstream tasks could be used as a proxy metric for evaluation, the main issue is that large language models are trained to be general / multi-purpose models where the space of downstream tasks is large and unknown during the time of pretraining. Thus from a fundamental standpoint, it is beneficial to have an indicator that is agnostic to downstream tasks. This is often why the standard way to compare language models is with pre-training perplexity (e.g., the training logs of OPT [1], GPT-3).
>
> Moreover, understanding the implicit biases needed for downstream performance may also lead to better training methods (instead of better evaluation methods) that might encourage the correct biases more strongly.
>
>
>
>
>
> >**Q3** 8JLN notes that this paper does not demonstrate if the insight gleaned in this work can lead to additional “flatness regularization” that induces better downstream performance on real datasets.
>
> We agree that experiments using flatness regularization on real datasets could be an interesting direction - however, our main goal is to show that implicit bias even has a role in downstream performance, because the existence of implicit bias for language models was not even clear. Thus, we leave explicit regularization in language modeling to future work. Previous works [2] show explicit flatness regularization with SAM can boost downstream performance when applying to downstream tasks themselves and the intermediate stages between pre-training and fine-tuning, but they did not study this on pre-training, partly because SAM is not efficient enough for pre-training (requires back prop for 2 times per step, and more steps to reach the same level of pre-training loss).
>
> [1] [OPT training logs](https://github.com/facebookresearch/metaseq/blob/main/projects/OPT/chronicles/OPT175B_Logbook.pdf)
> [2] Sharpness-Aware Minimization Improves Language Model Generalization.

---

> > ### Comment · Reviewer_8JLN · 2022-11-22
> > **Response**
> >
> > > While we focus on the saturation regime in the paper as a controlled way to compare models with the same pretraining loss, the overall takeaway is that the implicit bias of the training algorithms matters for downstream performance (no matter whether we are in the saturation regime or not).
> >
> > This overall takeaway seems speculative given that no experiments were done outside of the saturation regime.
> >
> > > While downstream tasks could be used as a proxy metric for evaluation, the main issue is that large language models are trained to be general / multi-purpose models where the space of downstream tasks is large and unknown during the time of pretraining. Thus from a fundamental standpoint, it is beneficial to have an indicator that is agnostic to downstream tasks. This is often why the standard way to compare language models is with pre-training perplexity (e.g., the training logs of OPT [1], GPT-3).
> >
> > That's fair.
> >
> > > We agree that experiments using flatness regularization on real datasets could be an interesting direction - however, our main goal is to show that implicit bias even has a role in downstream performance, because the existence of implicit bias for language models was not even clear.
> >
> > Sure. This goal is meaningful on its own but perhaps not that hard to achieve. It's widely known that optimizers and training procedures have their own inductive biases baked in which affect generalization. It's not surprising to see the same holds for LLMs in terms of pretraining vs finetuning performance.
> >
> > I stand by my original assessment and think that this work should be considered for acceptance.

---

> ### Author Response · Authors · 2022-11-18
> **Further Discussion**
>
> Dear reviewer, may we ask if you could respond to our comments? In our response below, we elaborated more on the practical implications of our findings. We incorporated these discussions in the revision (Section D). We also noted that previous works show an encouraging sign that explicit flatness regularization may also work in language modeling. Please let us know if you have other questions or concerns. Thank you!

---

### Official Review · Reviewer_6MPN · 2022-10-25

**Confidence:** 3
**Correctness:** 3
**Technical Novelty And Significance:** 3
**Empirical Novelty And Significance:** 3
**Recommendation:** 6

**Clarity, Quality, Novelty And Reproducibility:**

Clarity&Quality: The paper is clear, well-written and easy to follow.
Novelty: As far as I know, there is no existing paper discussing the same topic.
Reproducibility: The authors didn't provide the code. I think it's hard to reproduce the results by ourselves.

**Strength And Weaknesses:**

Strength:
- The paper is well-written and easy to follow.
- The paper is supported by many experiments to verify the claims that the authors proposed.
- Besides empirical results, the author also provided theoretical points of view to support their claim.

Weaknesses:
- The claim that flatness can decide the downstream performance is not well-supported. There are many factors that can affect the downstream performance, for example, the model size. One possible explanation is that the flatness could be a consequence of scaling up the model size, and the good performance is brought by the large model size, too. If that is the case, then flatness is not the reason for good downstream performance. The authors need to have more evidence to prove that flatness is the main reason that leads to good performance, otherwise, they may be both the consequence of another factor (like model size).
- When computing Hessian, I assume that you are using the loss of pretraining datasets, not downstream datasets. Then how can the flatness on the pretraining task reflect the situation on downstream tasks? This claim is not very intuitive. It may need more explanations.

**Summary Of The Paper:**

This paper addresses the issue of pre-training loss cannot fully explain downstream performance, they instead claim that the flatness of the model is well-correlated with downstream performance. The author showed that at the same level of pretraining loss, large models have better downstream performance, because of the flatness of large models. The authors conducted experiments with PCFG/HMM/OPT-generated data as downstream tasks. The authors also provided theories to support their claims.


**Summary Of The Review:**

The paper is good in terms of its qualities, but there are some problems that need to be answered to further prove the correctness of their claims. I would tend to accept this paper if they can give good answers to my questions. For now, I will give a borderline score.

---

> ### Author Response · Authors · 2022-11-11
> **Response to Reviewer 6MPN**
>
> Thanks for noting that “the paper is supported by many experiments” and “no existing paper discussing the same topic”. We address the questions and concerns below.
>
> >**Q1:** “The claim that flatness can decide the downstream performance is not well-supported. There are many factors that can affect downstream performance, for example, the model size.”
>
>  We believe that there is a misunderstanding here on the main message of the paper. We only claimed that the pre-training validation loss does NOT decide the downstream performance. This is shown by varying many factors, e.g., training time, model size (as the reviewer mentioned), while keeping the pre-training loss the same. These experiments call for what are the additional factors from the training algorithms which affect (but _not necessarily decide_) the downstream performance. These missing factors are called implicit bias of the algorithms, which is a well studied area in supervised learning [1,2,3,4,5,6]. After establishing that they are such missing factors, we study what could be likely the  *common* missing factors or implicit bias (as opposed to attributing to different factors for different scenarios).
>
> We show that flatness is likely *one kind* of implicit bias , because we empirically (1)show that model size, training time, and the pre-training algorithm's choice all affect flatness, and flatness in turn correlates well with the downstream performance, and (2) theoretically SGD prefers flat local minima.
>
> In other words, the implicit bias serves as the **interplay** between the factors of the pre-training algorithms and the downstream performance.
>
>
>
>
> >**Q2:** “How can the flatness on the pre-training task reflect the situation on downstream tasks?”
>
> We would like to first emphasize that to some extent, it might be somewhat surprising to see that flatness correlate well with the downstream perf (given the pretraining loss is the same). Identifying flatness as a common factor, and demonstrating such a correlation empirically are indeed the main contributions of the paper.
>
> Such a phenomenon can be intuitively explained by the fact that flatness encourages simpler models which tends to use more fundamental and transferable features. However, we note that a full theoretical explanation is very challenging—it still remains an open question why flatter models have better generalization for supervised learning despite that researchers generally believed that the key implicit bias of SGD is the flatness in the supervised setting [4,5]. Thus it’s even more challenging in the pre-training setting with the current theoretical tools. Therefore, we empirically show the correlation between flatness and downstream performance, leaving the theoretical explanation as an important open question. To make an initial step toward this direction, in Section 5, we show on the synthetic language that there are multiple solutions with the optimal pretraining loss but the flatter solution generalizes better to downstream tasks (Theorem 5.1). We hope that our work encourages the field to look further into the relationship between flatness (and other possible implicit biases) and downstream performance.
>
>
> >**Q3:** The authors didn't provide the code. I think it's hard to reproduce the results by ourselves.
>
> We believe reviewer 6MPN missed our submitted code, which was provided in the supplementary material. We will also make it publicly available.
>
> [1] [On the Implicit Bias in Deep-Learning Algorithms](https://arxiv.org/abs/2208.12591) (A survey)
> [2] Implicit bias of gradient descent on linear convolutional networks Neurips 2018
> [3] Gradient descent maximizes the margin of homogeneous neural networks. ICLR 2019
> [4] Label noise sgd provably prefers flat global minimizers. Neurips 2021
> [5] What Happens after SGD Reaches Zero Loss? --A Mathematical Framework ICLR 2022
> [6] Kernel and rich regimes in overparametrized models. COLT 2020

---

> ### Author Response · Authors · 2022-11-18
> **Further Discussion**
>
> Dear reviewer, may we ask if you could respond to our comments? In our response below, we explain in more detail how the implicit bias in pre-training can affect the downstream performance. Please let us know if you have other questions or concerns. Thank you!

---

> > ### Comment · Reviewer_6MPN · 2022-11-21
> > **Thanks for the reply**
> >
> > Thanks for your reply. I agree that it may be my misunderstanding. The message of this paper is flatness is one kind of implicit bias that can affect the downstream performance, but it does not decide the performance completely. This sounds more reasonable to me. And it's my fault to miss your submitted code.
> >
> > However, I think the contribution of this paper is limited as it just claims that flatness on the pre-training task reflects the situation on downstream tasks based on empirical results. In my opinion, if there is a considerable difference between the pre-training task and the downstream tasks, this claim may not able to hold.
> >
> > Anyways, I can increase my score from 5 to 6 as you pointed out my misunderstanding.

---

### Author Response · Authors · 2022-11-18
**General Response**

We thank all the reviewers for the detailed reviews. The reviewers noted that “there is no existing paper discussing the same topic”, “the investigation is both well-motivated and nicely executed” and “supported by many experiments”, and we presented "interesting empirical observations and theoretical results".

We addressed all the concerns in the individual comments. We list a summary of the changes we made to the paper here:

**Detailed practional implications in Section D.**
We expanded Section D text to make it clearer how the findings in this paper may help language models in practice: (1) designing pre-training algorithms to use better implicit bias for transferability to downstream tasks (2) more reliable metrics for language models pre-training and (3) potential explicit regularization algorithms.


**More detailed description of viewing a small transformer as a large transformer in Section B.1.** We add a more detailed justification of "the smaller transformer architecture is a subset of the larger transformer architecture family" in Section B.1. We also attach the code to find a large transformer with the same functionality as a small transformer in [this link](https://anonymous.4open.science/r/iclr2023_2440-DFBD/from_small_to_large_add_layers.ipynb).

**Experiments with 15% mask rate in Section E.** During rebuttals, we added experiments of pre-training with 15% mask rate and and evaluating with one mask per sentence. Results are presented in Section E. The conclusions of Section 2 and Section 4 still hold.

---

### Decision · Program_Chairs · 2023-01-20

**Decision:**

Reject

**Justification For Why Not Higher Score:**

While this work tackles an interesting problem and raises an interesting finding, without theoretical backing it's unclear if this finding generalizes to other downstream tasks. Interventions on model flatness and theoretical analyses of flatness on a wider range of tasks would make this a stronger paper.

**Justification For Why Not Lower Score:**

N/A

**Metareview: Summary, Strengths And Weaknesses:**

This work shows that pretraining loss does not fully explain downstream performance and claim that the flatness of the model is well-correlated with downstream performance. The authors conduct experiments with synthetically generated data as downstream tasks and tie that together with theories to show how flatness improves downstream performance.

Strengths:
* The paper is well written.
* This work tackles an important question in representation learning, which is the question of how pretraining performance affects downstream performance.
* The work found a correlation between the decay of the trace of the Hessian matrix evaluated on the training data and the improved performance on downstream tasks.
* The work provides some theoretical analysis that connects flatter solutions to the pretraining loss and generalization performance of downstream tasks.
* This could be a new metric for language models other than perplexity.

Weaknesses:
* The work is unable to show a causal link between flatness and downstream performance. In particular, the work does not attempt to perform any interventions, such as regularising the trace of the Hessian or using the SAM optimizer to try to improve the flatness directly. This means the common causal factor for flatness and downstream performance could still be things such as the model size.
* It's unclear if this link between flatness and downstream performance generalise to more tasks.
* The theoretical analysis is too constrained for general practical applicability.
* The work shows that flatter minima leads to better generalization but only in the special case of the Dyck language experiment.

**Summary Of Ac-Reviewer Meeting:**

This paper was a borderline paper and so was discussed in a hybrid physical/virtual meeting. Reviewers raised high-level concerns that the relationship claimed in the paper might not generalize to other downstream tasks given the lack of theoretical backing. The reviewers agreed that the work had solid contributions including bringing in a new metric other than perplexity and showing that flatness is a good measure that correlates with downstream performance. However, overall the reviewers felt that the lack of interventions limited the applicability of the paper and cast doubt on its generalizability. In a future revision, it would be helpful to show how improving flatness during pretraining affects metrics.